# RGGT: A Generative-Prior-Guided Transformer for Unified Rigid and Non-Rigid Point Cloud Registration

**Chengyu Zheng** [1]  **Songlin Yang** [2]  **Jin Huang** [3]  **Honghua Chen** [4]  **Weiming Wang** [5]  **Haoran Xie** [4]  **Fu Lee Wang** [5]  **Mingqiang Wei** [1]

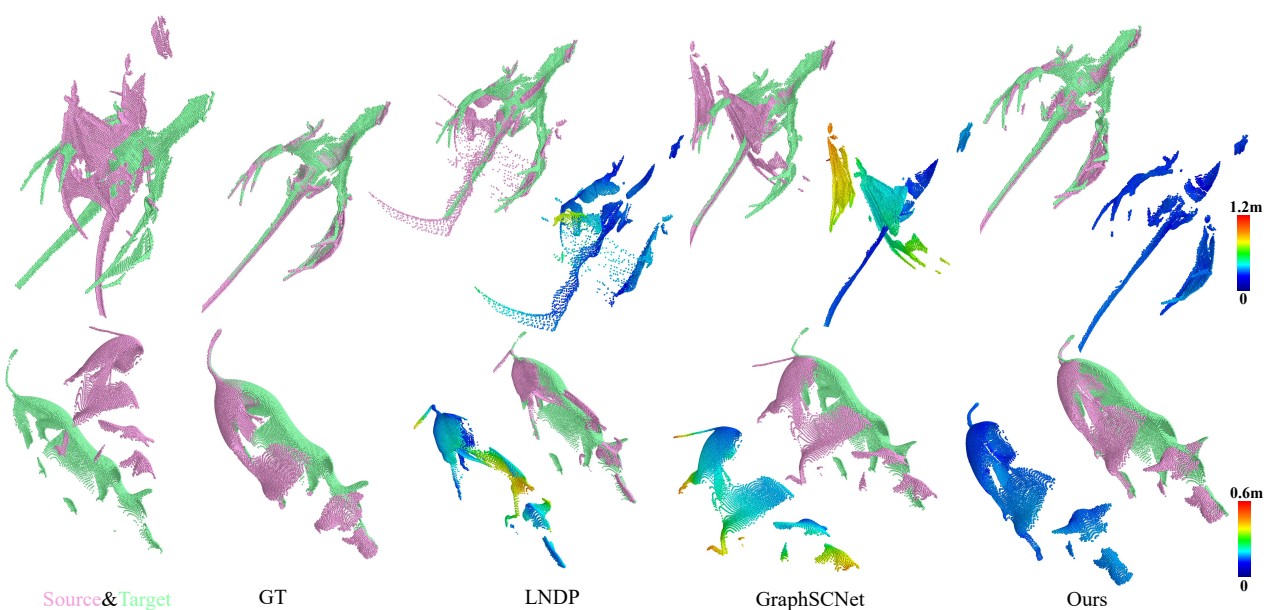

*Figure 1.* Qualitative comparison of state-of-the-art non-rigid registration methods on 4DMatch (top) and 4DLoMatch (bottom). The leftmost column shows the source (pink) and target (green) point clouds. For each method, the left visualization depicts the deformed source aligned to the target, while the right visualization shows the per-point error map of the deformed source with respect to the ground-truth source. RGGT achieves smoother and more accurate registration on 4DMatch, and maintains superior robustness in low-overlap scenarios on 4DLoMatch, exhibiting fewer misalignments and lower errors compared to LNDP and GraphSCNet.

## Abstract

Point cloud registration can be categorized into rigid and non-rigid settings depending on the motion characteristics of the underlying objects. Rigid alignment assumes a single global transformation under which corresponding points remain geometrically consistent across scales, whereas non-rigid alignment involves spatially varying deformations, where geometric similarity holds only locally and semantic correspondence dominates at larger scales. This multi-scale discrepancy creates an optimization gap that has made unified registration particularly challenging. To this end, we propose RGGT, a Generative-Prior-Guided Transformer that unifies rigid and non-rigid registration within a shared optimization space. Through coordinated design at the representation, architecture, and supervision levels, RGGT jointly captures local geometric details and global structural semantics: generative priors enrich point features with unified geometric-semantic cues; a Global-Self-Cross Attention module models long-range structure, local interaction, and bidirectional cross-shape reasoning; and a dual correspondence-reconstruction objective provides consistent su-

[1]Nanjing University of Aeronautics and Astronautics, Nanjing, China [2]MMLab@HKUST, The Hong Kong University of Science and Technology, Hong Kong SAR, China [3]Hong Kong Applied Science and Technology Research Institute, Hong Kong SAR, China [4]School of Data Science, Lingnan University, Hong Kong SAR, China [5]School of Science and Technology, Hong Kong Metropolitan University, Hong Kong SAR, China. Correspondence to: Mingqiang Wei <mqwei@nuaa.edu.cn>.

*Proceedings of the $43^{rd}$ International Conference on Machine Learning*, Seoul, South Korea. PMLR 306, 2026. Copyright 2026 by the author(s).

pervision for both deformation types. Extensive experiments on rigid (ModelNet40, 3DMatch, KITTI) and non-rigid (4DMatch) benchmarks demonstrate that RGGT achieves state-of-the-art accuracy across both rigid and non-rigid settings within a single unified framework. Code is available at https://github.com/zhengcy-lambo/RGGT.

## 1. Introduction

Point cloud registration aims to align two sets of 3D points by estimating their spatial transformation. According to the motion characteristics of the underlying objects, registration can be categorized as *rigid* or *non-rigid*. Rigid registration estimates a single global transformation that aligns overlapping regions without deformation, while non-rigid registration models spatially varying deformations where local parts may move independently. As a fundamental problem in 3D vision, registration serves as the foundation for 3D reconstruction (Yu et al., 2020), scene understanding (Zhou et al., 2019), and robotic manipulation (Paravati et al., 2017).

Although rigid and non-rigid registration differ in their deformation characteristics, their formulations share a common paradigm: both can be expressed as estimating point-wise correspondences (Huang et al., 2021; Yu et al., 2021; Qin et al., 2022; Li & Harada, 2022b; Qin et al., 2023; Wang et al., 2025) or minimizing reconstruction-based objectives (Yew & Lee, 2022; Chen et al., 2023). This indicates the potential for a unified framework. However, no existing method has achieved robust and consistent performance across both tasks. The difficulty arises from a *multi-scale discrepancy*: rigid registration relies on stable cross-scale geometric similarity, whereas non-rigid registration must reason over locally coherent but globally semantically driven correspondences. This discrepancy leads to optimization conflicts and exposes a representational gap in existing architectures, which lack mechanisms to capture fine-grained geometry and global semantics jointly.

Recent advances in 2D correspondence learning offer an important insight for unified registration. Diffusion-based vision models have been shown to provide strong zero-shot multi-scale correspondences despite never being trained for matching tasks (Tang et al., 2023; Jiang et al., 2024; Song et al., 2025). These results reveal a key property of diffusion latent spaces: they naturally encode multi-scale geometric and semantic cues that remain stable under large appearance or structural variations. This observation motivates us to explore whether similar deformation-robust properties arise in 3D diffusion-based generative models. In particular, we find that TRELLIS (Xiang et al., 2025), a large-scale 3D generative model trained on diverse shape corpora, pro-

duces latent voxel embeddings that exhibit precisely the characteristics needed for unified registration: they retain fine-grained geometric detail while maintaining globally coherent structural semantics across shapes and deformation patterns. Moreover, unlike 2D vision priors or projection-based depth features (Jiang et al., 2025; Wang et al., 2023; Zheng et al., 2025), TRELLIS learns geometry directly in 3D space, avoiding cross-modal domain gaps and providing deformation-invariant structural priors compatible with both rigid and non-rigid alignment.

Based on the above observations, we propose RGGT, a Generative-Prior-Guided Transformer that unifies rigid and non-rigid registration within a shared optimization space. RGGT achieves this through coordinated design across the representation, architecture, and supervision levels.

At the representation level, generative priors distilled from TRELLIS (Xiang et al., 2025) enrich point features with unified geometric-semantic cues, preserving global structural coherence while allowing local flexibility. Here, "geometric–semantic" means that the features may reflect both local geometric structure and semantic consistency, rather than implying an explicit disentanglement between geometric and semantic components. At the architecture level, we introduce a Global-Self-Cross Attention module that jointly models long-range structural context, local geometric interactions, and bidirectional cross-shape information flow, enabling reliable correspondence reasoning. At the supervision level, we design a dual correspondence-reconstruction objective that provides consistent feature-level and coordinate-level supervision, enforcing holistic geometric consistency and stabilizing optimization across deformation types. Through these designs, RGGT effectively unifies the rigid and non-rigid transformations. Our main contributions are summarized as follows:

- We propose RGGT, the first framework that unifies rigid and non-rigid point cloud registration within a shared optimization space.

- We incorporate geometric-semantic priors distilled from native 3D generative models, providing globally coherent yet locally flexible structural cues that benefit both rigid consistency and non-rigid deformation modeling.

- We introduce a dual correspondence-reconstruction objective that enforces both local matching fidelity and global geometric consistency, yielding stable optimization across deformation types.

- Extensive results show that RGGT achieves state-of-the-art performance on both rigid and non-rigid benchmarks under a single unified framework, validating the robustness and general applicability of our approach.

## 2. Related Work

### 2.1. Rigid Point Cloud Registration

Rigid point cloud registration aims to estimate a single transformation that globally aligns two partially overlapping 3D point clouds. Traditional approaches (Besl & McKay, 1992; Rusu et al., 2008; Sun et al., 2009; Rusu et al., 2009; Salti et al., 2014; Zhang et al., 2021) commonly rely on hand-crafted geometric descriptors to encode local structural information. However, these features often limit the generalization and robustness, particularly in challenging scenarios characterized by low overlap.

Recent advances in learning-based 3D registration (Huang et al., 2021; Yew & Lee, 2022; Ao et al., 2023; Mei et al., 2023; Liu et al., 2023; Zheng et al., 2025) have rapidly progressed from fully convolutional neural network like FCGF (Choy et al., 2019), which computes the features without relying on keypoint detection, to dominant transformer-based architectures. REGTR (Yew & Lee, 2022) employs attention to predict point correspondences and enable end-to-end rigid transformation estimation. Subsequent works refined this paradigm: GeoTransformer (Qin et al., 2022) embeds the geometric features of point clouds within the Transformer for robust superpoint matching. while UGP (Zeng et al., 2025) adapts the architecture for LiDAR point cloud generalization using BEV features and attention pruning.

However, all existing methods are fundamentally limited by the rigid-only assumption: they model a single global transformation and therefore lack mechanisms to capture locally varying geometric changes. Such a formulation prevents these approaches from maintaining consistent optimization behavior when extended to non-rigid scenarios, where local motions violate global rigidity. In contrast, our unified framework leverages generative-prior-guided geometric-semantic representations and a dual correspondence-reconstruction objective to support stable optimization across both rigid and non-rigid deformation types.

### 2.2. Non-rigid Point Cloud Registration

Non-rigid point cloud registration aims to find a deformation function—parameterized, for instance, by dense displacement fields (Li & Harada, 2022a), local affine transformations (Li & Harada, 2022b), or embedded deformation graphs (Sumner et al., 2007)—to align a source point cloud with a target. State-of-the-art methods have leveraged deep neural networks to model these deformations, employing techniques such as hierarchical motion decomposition (Li & Harada, 2022b), graph-based spatial consistency (Qin et al., 2023), and semantic feature injection (Chen et al., 2025). Recent AniSym-Net (Wang et al., 2025) leverages anisotropic shape-motion fields with symplectic constraints

to better handle occlusions and near-isometric deformations. However, most non-rigid registration methods estimate local deformation fields independently across regions, which often leads to overfitting, structural distortions, or the loss of globally coherent shape behavior. Their reliance on purely local deformation modeling makes it difficult to maintain consistent optimization when integrating with rigid registration, whose global transformation assumptions fundamentally differ. In contrast, our approach introduces unified geometric-semantic representations and shared optimization objectives that impose global structural coherence, enabling stable learning across both rigid and non-rigid deformation types.

### 2.3. Pre-trained Foundation Model for Registration

The success of large-scale pre-trained models in 2D vision has spurred their application to 3D point cloud registration. A common paradigm involves transferring knowledge from 2D foundation models to establish robust correspondences. This has been demonstrated for zero-shot rigid registration by leveraging features from pre-trained foundation models (Jiang et al., 2025; Wang et al., 2023; Zheng et al., 2025) and for non-rigid registration by injecting semantic priors (Chen et al., 2025). Other works utilize pre-trained models to bridge modalities, such as for image-to-point cloud alignment (Wang et al., 2024), or to learn generalizable 3D representations from 2D knowledge (Zhang et al., 2023a). A fundamental challenge is that transferring features from 2D image-based models to 3D geometric data often results in a loss of fine-grained structural information and a domain gap. To address this, we employ a native 3D generative model as the geometry encoder, extracting features that are inherently rich in 3D structural information, providing a consistent and robust representation for both rigid and non-rigid alignment.

## 3. Method

### 3.1. Problem Definition

Given a source point cloud $P$ and a target point cloud $Q$, rigid registration estimates a single transformation $(R, t)$, where $R \in SO(3)$ and $t \in \mathbb{R}^3$, that aligns $P$ to $Q$. In contrast, non-rigid registration aims to recover a continuous warping function $\mathcal{W} : \mathbb{R}^3 \rightarrow \mathbb{R}^3$ that maps each point in $P$ to its counterpart in $Q$ by modeling spatially varying local deformations. Despite their different motion assumptions, both tasks can be formulated as learning (1) point-wise correspondences between two point sets (Huang et al., 2021; Li & Harada, 2022b), and (2) coordinate-level reconstruction that enforces global geometric consistency. Accordingly, we express the overall registration as a unified mapping:

$$(\tilde{P}, \tilde{Q}, F_P, F_Q) = \Phi(P, Q, F_P^{tre}, F_Q^{tre}), \qquad (1)$$

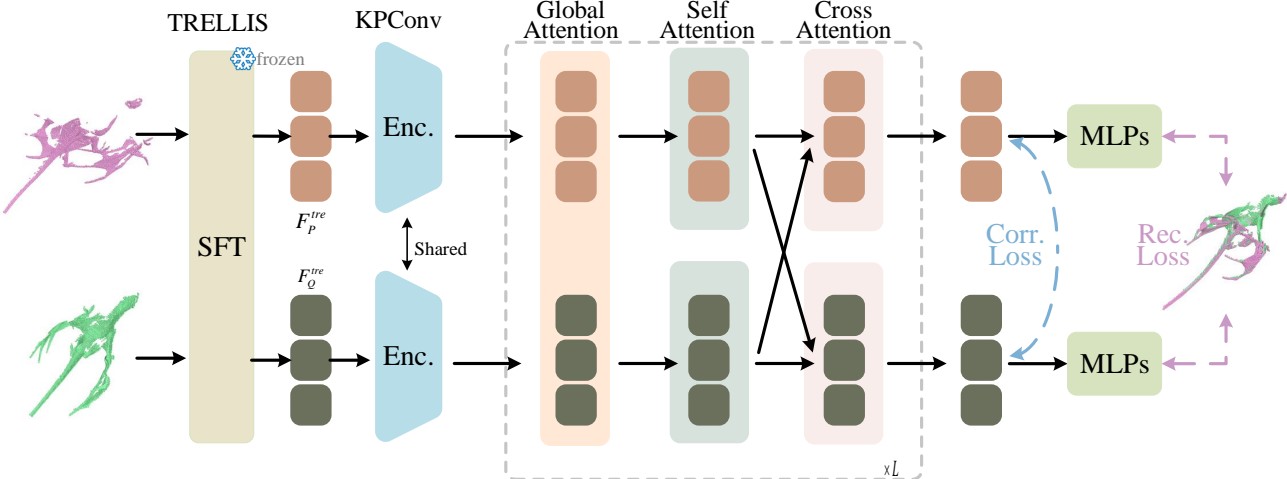

*Figure 2.* Overview of RGGT. We first extract generative-prior features from both source and target point clouds using the second-stage Sparse Flow Transformer (SFT) of TRELLIS, which provides structure-aware geometric-semantic representations. These features are encoded by KPConv and fed into a Transformer equipped with Global, Self, and Cross Attention layers to jointly capture global context, local geometry, and cross-shape interactions. The resulting features are directly used for feature-level contrastive correspondence learning, while two-layer MLP heads regress the reconstructed coordinates of each point cloud in the other's frame. The network is supervised by two complementary objectives: a Correspondence Loss, applied on transformer features to encourage discriminative point-wise matching, and a Reconstruction Loss, which enforces coordinate-level geometric consistency across deformation types.

where $\Phi$ denotes our unified registration function and $F_P^{tre}, F_Q^{tre}$ are features obtained from TRELLIS (Xiang et al., 2025). The network outputs (1) correspondence-aware features $F_P$ and $F_Q$, trained contrastively to produce high similarity for matching points and low similarity for non-matching pairs, and (2) $\tilde{P}$ and $\tilde{Q}$, representing the source and target point clouds reconstructed in each other's co-ordinate frame. This joint correspondence-reconstruction formulation provides the foundation for our unified optimization across both rigid and non-rigid registration.

Fig. 2 summarizes the RGGT architecture, which consists of three main components: (1) a generative-prior-guided feature extractor that produces unified geometric–semantic representations, capturing both fine-grained local geometry and globally coherent structure; (2) a transformer backbone that models rigid and non-rigid transformations within a shared feature space; and (3) a feature-matching and shape-reconstruction module that provides consistent feature-level and coordinate-level supervision across deformation types. We describe each component in detail below.

### 3.2. Feature Encoding

Recent studies, such as ZeroReg (Wang et al., 2023), Zero-Match (Jiang et al., 2025), RARE (Zheng et al., 2025), and DV-Matcher (Chen et al., 2025), leverage pre-trained 2D visual models to guide point cloud registration in both rigid and non-rigid settings. However, these approaches suffer from inherent modality limitations. For RGB-D-based methods (e.g., ZeroReg and ZeroMatch), their dependence on

RGB imagery limits applicability in scenarios lacking color information. For projection-based methods (e.g., RARE and DV-Matcher), projecting 3D point clouds onto depth maps inevitably causes geometric information loss due to spatial overlap and occlusion. In addition, most pre-trained 2D foundation models are trained solely on RGB data, further exacerbating the modality discrepancy. To overcome these issues, we adopt TRELLIS (Xiang et al., 2025) (text-to-3D version) as our feature encoder to extract generative-prior features directly from raw point clouds, providing unified geometric-semantic embeddings for both rigid and non-rigid registration.

Specifically, we leverage the second-stage SFT model of TRELLIS to extract geometry-aware latent features. Given a voxelized point cloud $P$, the SFT takes the voxel coordinates as input and uses the shape name as a text condition, enabling the model to generate semantically grounded geometric embeddings. We extract features from the last two upsampling layers of the SFT decoder and concatenate them to form the final TRELLIS feature representation. Formally, the encoded generative-prior feature is defined as

$$F_P^{tre} = \Psi_{SFT}(P, T), \qquad (2)$$

where $\Psi_{SFT}(\cdot)$ denotes the SFT feature extractor conditioned on point positions and text prompt $T$ (shape name), and $F_P^{tre}$ represents the concatenated multi-scale voxel embeddings capturing both fine geometric details and high-level semantic context. In our implementation, the text prompt $T$ only provides a coarse category-level cue, rather than a fine-grained geometric description. Since the SFT is

strongly conditioned on the voxelized 3D input itself, most geometric information is carried by the input point cloud and the resulting TRELLIS prior features. Therefore, the text prompt serves mainly as a weak semantic cue rather than a necessary source. These features are fused with the raw 3D coordinates and subsequently fed into our transformer backbone for joint rigid and non-rigid registration.

To assess whether TRELLIS features indeed encode meaningful correspondences, we perform a Principal Component Analysis (PCA) on the extracted voxel-structured embeddings, as shown in Fig. 3. Color variations reflect feature variations, where similar colors indicate semantically or geometrically consistent regions. The visualization shows that corresponding regions in the source and target point clouds exhibit nearly identical color patterns in both non-rigid (top) and rigid (bottom) examples, indicating that TRELLIS preserves both local geometric structure and global semantic coherence. Such properties make TRELLIS features well-suited for robust correspondence learning under both rigid and non-rigid deformation. Beyond the qualitative PCA visualization in the figure, we further quantify the registration relevance of TRELLIS features by direct nearest-neighbor matching in the feature space, with results provided in Appendix G.

### 3.3. Transformer Backbone

Since directly feeding raw point clouds into a transformer is memory-intensive, we first employ KPConv (Thomas et al., 2019) as a lightweight point encoder to extract features and downsample the input. KPConv applies ResNet-style convolutional blocks and strided kernels to progressively reduce the input point clouds into a compact set of keypoints $P'$ and $Q'$, along with their associated features $F_{P'}^{tre}$ and $F_{Q'}^{tre}$. The downsampled features extracted by KPConv are first linearly projected into a lower-dimensional embedding space and then passed through a stack of $L = 6$ transformer encoder layers. Unlike prior self-cross attention designs such as GeoTransformer (Qin et al., 2022) and REGTR (Yew & Lee, 2022), our transformer adopts a Global–Self–Cross Attention design. Specifically, it introduces an additional global attention branch before local self- and cross-attention to jointly aggregate source-target structural context. This allows the network to reason about the global shape layout before establishing fine-grained local correspondences. Each encoder layer therefore consists of three attention components designed to jointly model local geometry, global structure, and cross-shape correspondence: (1) Global Attention: captures long-range structural dependencies between the source and target point clouds, providing global context that helps distinguish rigid and non-rigid transformations. (2) Self Attention: operates independently on each point cloud to model local geometric relationships and maintain structural consistency within each shape. (3) Cross

Attention: enables bidirectional information exchange between the two point clouds, allowing each point to attend to potential correspondences in the other cloud for accurate alignment reasoning. Formally, given the source and target keypoint features $F_{P'}$ and $F_{Q'}$, the three attention components differ only in how the queries ($Q$), keys ($K$), and values ($V$) are constructed:

$$\text{Global-Attn: } [Q, K, V] = [F_{P'}\|F_{Q'}, F_{P'}\|F_{Q'}, F_{P'}\|F_{Q'}], \tag{3}$$

$$\text{Self-Attn: } [Q, K, V] = [F_{P'}, F_{P'}, F_{P'}] \text{ or } [F_{Q'}, F_{Q'}, F_{Q'}], \tag{4}$$

$$\text{Cross-Attn: } [Q, K, V] = [F_{P'}, F_{Q'}, F_{Q'}] \text{ or } [F_{Q'}, F_{P'}, F_{P'}], \tag{5}$$

where $\|$ denotes feature concatenation. This unified formulation highlights that global attention integrates both point clouds to capture long-range structural dependencies, self-attention models intra-cloud geometric relations, and cross-attention enables inter-cloud correspondence reasoning. Note also that the outputs of all heads are concatenated and linearly projected to form the final feature.

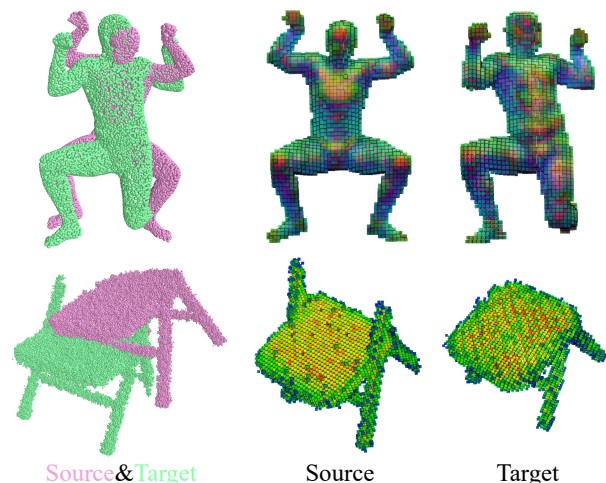

*Figure 3.* Visualization of PCA-projected TRELLIS features. The top row shows a non-rigid example and the bottom row a rigid one. Similar color patterns across source and target indicate that TRELLIS produces semantically and geometrically coherent feature distributions, even under deformation or partial overlap.

### 3.4. Prediction Objectives

Based on the final feature representations produced by the transformer backbone, we introduce two prediction objectives: a correspondence objective that supervises feature-level matching, and a reconstruction objective that provides coordinate-level geometric supervision. Together, this dual correspondence-reconstruction design enforces both discriminative correspondence reasoning and globally coherent

geometric alignment, forming the foundation of our unified optimization across rigid and non-rigid settings.

**Correspondence Loss.** To promote geometry-aware correspondence learning, we adopt the InfoNCE loss (van den Oord et al., 2018) on the learned point features. For a point $p \in P'$ with a correspondence in $Q'$, the loss for the source cloud is formulated as

$$\mathcal{L}_f^P = -\mathbb{E}_{p \in P'} \left[ \log \frac{f(p, pos_p)}{f(p, pos_p) + \sum_{neg_p} f(p, neg_p)} \right],$$
(6)

where $f(\cdot, \cdot)$ is a log-bilinear similarity function defined as $f(p, c) = \exp(\bar{f}_p^\top W_f \bar{f}_c)$, and $\bar{f}_p$ denotes the transformer feature of point $p$. $pos_p$ and $neg_p$ indicate positive and negative keypoints in $Q'$, respectively. They are determined using positive and negative margins $(r_p, r_n)$ set to $(m, 2m)$, where $m$ is the voxel size in the last KPConv layer. All points outside the negative margin are used as negative samples. To maintain symmetry between the two point clouds, the learnable transformation matrix $W_f$ is constrained to be symmetric. This is because $\bar{f}_p$ and $\bar{f}_c$ are extracted by the same TRELLIS encoder and therefore lie in the same feature space. We parameterize $W_f$ as the sum of an upper-triangular matrix $U_f$ and its transpose, i.e., $W_f = U_f + U_f^\top$, which reduces redundant parameters. The same formulation applies symmetrically for the target cloud $Q$. Generally, this contrastive objective maximizes similarity between corresponding points while minimizing it for non-matching pairs, producing correspondence-discriminative embeddings.

**Reconstruction Loss.** Complementary to feature-level correspondence supervision, we further utilize the final feature representation to directly predict transformed keypoint coordinates. A two-layer MLP is employed to regress these coordinates:

$$\tilde{P}' = \text{ReLU}(\bar{f}_p W_1 + b_1) W_2 + b_2,$$
(7)

where $\tilde{P}'$ denotes the reconstruction of $P'$ in $Q'$'s coordinate frame, and vice versa for $\tilde{Q}'$. $W_1$, $W_2$, $b_1$, and $b_2$ are learnable weights and biases. We supervise the predicted coordinates using an $\ell_1$ loss:

$$\mathcal{L}_c^P = \frac{1}{M'} \sum_{i=1}^{M'} \left\| p_i^{gt} - \tilde{p}_i' \right\|_1,$$
(8)

where $M'$ is the number of keypoints and $p_i^{gt}$ denotes the ground-truth position of $\tilde{p}_i'$ in $Q'$'s coordinate frame. A symmetric loss $\mathcal{L}_c^Q$ is applied in the opposite direction. This reconstruction objective reinforces spatial structure preservation and provides coordinate-level supervision that complements the feature-level correspondence loss, together improving the geometric consistency and robustness of registration. Finally, the overall training objective combines both correspondence and reconstruction losses as

$$\mathcal{L} = \lambda \mathcal{L}_f^P + \lambda \mathcal{L}_f^Q + \mathcal{L}_c^P + \mathcal{L}_c^Q,$$
(9)

where $\lambda = 0.1$ are weighting coefficients that balance the feature-level and coordinate-level supervision for the two point clouds.

## 4. Experiments

**Implementation details.** We train our network using the AdamW (Loshchilov & Hutter, 2017) optimizer with an initial learning rate of $1 \times 10^{-4}$ and a weight decay of $1 \times 10^{-6}$. Training is performed in a mixed manner across the four datasets 4DMatch, 3DMatch, KITTI, and ModelNet40 to enhance the generalization capability. All experiments are conducted on a single NVIDIA RTX 3090 GPU.

### 4.1. Metrics

Following prior works (Qin et al., 2023; Huang et al., 2021), we evaluate the performance using both non-rigid and rigid registration metrics.

**Non-rigid metrics.** We adopt four standard evaluation metrics: (1) 3D End-Point Error (EPE) is the average distance between warped points under the estimated and ground-truth warp functions, measured in meters (m); (2) 3D Accuracy (Strict, AccS) is the fraction of points whose EPEs are below 2.5 or relative errors below 2.5%; (3) 3D Accuracy (Relaxed, AccR) is the fraction of points whose EPEs are below 5 or relative errors below 5%; and (4) Outlier Ratio (OR) is the fraction of points whose relative errors exceed 30%.

**Rigid metrics.** We additionally measure Relative Rotation Error (RRE) and Relative Translation Error (RTE) on all point clouds between the registered scans. Lower values across all metrics indicate better alignment accuracy.

### 4.2. Evaluations on Non-rigid Dataset

**Datasets.** We evaluate our method on two non-rigid datasets: 4DMatch and 4DLoMatch. 4DMatch (Li & Harada, 2022a) is a challenging benchmark for non-rigid point cloud registration, constructed from animation sequences in the DeformingThings4D dataset (Li et al., 2021b). It contains 1,232 sequences for training, 176 for validation, and 353 for testing. The test set is further divided into two subsets, 4DMatch and 4DLoMatch, according to the overlap ratio between paired point clouds, with a threshold of 45%. 4DMatch includes pairs with sufficient overlap, while 4DLoMatch focuses on low-overlap cases that are particularly challenging for correspondence estimation.

**Quantitative comparisons.** We compare RGGT against state-of-the-art methods for non-rigid registration and scene flow estimation, including NSFP (Li et al., 2021a), Ner-

*Table 1.* Comparisons with state-of-the-art methods on 4DMatch and 4DLoMatch. Bold numbers indicate the best performance, while underlined numbers denote the second best. RGGT consistently outperforms prior approaches across all metrics, achieving the lowest EPE and OR and the highest AccS and AccR scores.

| | 4DMatch | | | | 4DLoMatch | | | |
|---|---|---|---|---|---|---|---|---|
| | EPE↓ | AccS↑ | AccR↑ | OR↓ | EPE↓ | AccS↑ | AccR↑ | OR↓ |
| NSFP (Li et al., 2021a) | 0.265 | 8.7 | 18.7 | 65.0 | 0.495 | 0.4 | 1.6 | 84.8 |
| Nerfiles (Park et al., 2021) | 0.280 | 12.7 | 25.4 | 58.9 | 0.498 | 1.1 | 3.0 | 82.2 |
| PointPWC-Net (Wu et al., 2020) | 0.182 | 6.3 | 21.5 | 52.1 | 0.279 | 1.7 | 8.2 | 55.7 |
| FLOT (Puy et al., 2020) | 0.133 | 7.7 | 27.2 | 40.5 | 0.210 | 2.7 | 13.1 | 42.5 |
| DGFM (Donati et al., 2020) | 0.152 | 12.3 | 32.6 | 37.9 | 0.148 | 1.9 | 6.2 | 64.6 |
| SyNoRiM (Huang et al., 2023) | 0.099 | 22.9 | 49.9 | 26.0 | 0.170 | 10.6 | 30.2 | 31.1 |
| Lepard+NICP (Li & Harada, 2022a) | 0.097 | 51.9 | 65.3 | 23.0 | 0.283 | 16.8 | 26.4 | 53.0 |
| LNDP (Li & Harada, 2022b) | 0.078 | 61.1 | 73.8 | 17.6 | 0.180 | 26.4 | 40.8 | 34.1 |
| GraphSCNet (Qin et al., 2023) | 0.043 | 72.3 | 84.4 | 9.4 | 0.121 | 41.0 | 58.2 | 21.0 |
| Ours | **0.031** | **73.9** | **89.7** | **7.1** | **0.052** | **55.0** | **79.0** | **6.8** |

*Table 2.* Comparison with state-of-the-art methods on the rigid registration benchmark. Non-rigid methods are evaluated under a unified SVD-based rigid post-processing. Bold numbers indicate the best performance, and underlined numbers indicate the second best. RGGT achieves the lowest error, demonstrating its effectiveness on rigid registration within a unified framework.

| | ModelNet | | 3DMatch | | 3DLoMatch | | Kitti | |
|---|---|---|---|---|---|---|---|---|
| | RRE↓ | RTE↓ | RRE↓ | RTE↓ | RRE↓ | RTE↓ | RRE↓ | RTE↓ |
| PointNetLK (Aoki et al., 2019) | 29.725 | 0.297 | 5.16 | 0.136 | 4.20 | 0.161 | 0.57 | 0.259 |
| OMNet (Xu et al., 2021) | 2.947 | 0.032 | 2.16 | 0.067 | 3.36 | 0.103 | 0.30 | 0.072 |
| DCP-v2 (Wang & Solomon, 2019) | 11.975 | 0.171 | 3.55 | 0.101 | 6.10 | 0.139 | 0.49 | 0.220 |
| RPM-Net (Yew & Lee, 2020) | 2.560 | 0.025 | 1.62 | 0.053 | 2.54 | 0.074 | 0.24 | 0.068 |
| Predator (Huang et al., 2021) | 2.700 | 0.027 | 2.03 | 0.064 | 3.05 | 0.093 | 0.24 | 0.056 |
| GeoTransformer (Qin et al., 2022) | 2.564 | 0.026 | 1.72 | 0.062 | 2.91 | 0.087 | 0.23 | 0.062 |
| MAC (Zhang et al., 2023b) | 2.048 | 0.023 | 1.60 | 0.052 | 2.57 | 0.075 | 0.39 | 0.128 |
| FastMAC (Zhang et al., 2024) | 2.275 | 0.026 | 1.86 | 0.065 | 2.92 | 0.085 | 0.38 | 0.120 |
| GraphSCNet (Qin et al., 2023) | 2.880 | 0.028 | 1.93 | 0.082 | 3.57 | 0.106 | 0.45 | 0.156 |
| PARE-Net (Yao et al., 2024) | / | / | 1.70 | 0.068 | 2.87 | 0.088 | 0.23 | 0.049 |
| ZeroReg (Wang et al., 2023) | / | / | 2.19 | 0.077 | 3.31 | 0.107 | / | / |
| ZeroMatch (Jiang et al., 2025) | / | / | 3.60 | 0.108 | / | / | / | / |
| BUFFER-X (Seo et al., 2025) | / | / | 1.79 | 0.057 | 3.21 | 0.094 | 0.27 | 0.077 |
| Ours | **1.452** | **0.013** | **1.55** | **0.043** | **2.52** | **0.065** | **0.21** | **0.043** |

fies (Park et al., 2021), PointPWC-Net (Wu et al., 2020), FLOT (Puy et al., 2020), DGFM (Donati et al., 2020), SyNoRiM (Huang et al., 2023), Lepard+NICP (Li & Harada, 2022a), LNDP (Li & Harada, 2022b), and GraphSCNet (Qin et al., 2023). As shown in Table 1, RGGT consistently outperforms all baselines on both the 4DMatch and 4DLoMatch benchmarks, demonstrating its ability to handle a wide range of non-rigid deformations. In particular, compared with the previous best method GraphSCNet, RGGT achieves significant gains of +1.6 and +14.0 percentage points (pp) in AccS, and +5.3 and +20.8 pp in AccR on 4DMatch and 4DLoMatch, respectively, highlighting its substantial improvement in accuracy and robustness. Although RGGT performs well on established low-overlap benchmarks, extremely low-overlap cases remain challenging when the shared geometric evidence becomes insufficient. We provide representative failure cases in Appendix I.

**Qualitative comparisons.** As shown in Figure 1, our method produces accurate and smooth alignments compared to existing approaches. Additional qualitative results, including detailed error maps and comparisons on 4DMatch and 4DLoMatch (e.g., *deer*, *rabbit*, and *dancer* examples), are provided in Appendix C.

### 4.3. Evaluations on Rigid Dataset

**Dataset.** We evaluate RGGT on four standard rigid registration benchmarks: ModelNet40 (Wu et al., 2015), 3DMatch (Zeng et al., 2017), 3DLoMatch, and KITTI (Geiger et al., 2012). ModelNet40 contains CAD models from 40 categories. Following (Qin et al., 2022), we generate partial-view pairs by uniformly sampling mesh surfaces, then applying random cropping and subsampling. For indoor scenes, we use 3DMatch and its low-overlap variant

3DLoMatch, comprising RGB-D scans from 62 scenes (8 for testing), with overlaps greater than 30% and between 10%-30%, respectively. For outdoor evaluation, we adopt KITTI's LiDAR sequences from urban driving scenarios and, following (Choy et al., 2019), use sequences 8 to 10 for testing.

**Quantitative comparisons.** We compare RGGT to rigid registration methods: PointNetLK (Aoki et al., 2019), OM-Net (Xu et al., 2021), DCP-v2 (Wang & Solomon, 2019), RPM-Net (Yew & Lee, 2020), Predator (Huang et al., 2021), GeoTransformer (Qin et al., 2022), MAC (Zhang et al., 2023b), FastMAC (Zhang et al., 2024), PARE-Net (Yao et al., 2024), BUFFER-X (Seo et al., 2025). For non-rigid baselines GraphSCNet (Qin et al., 2023) included in the rigid evaluation, we follow a unified post-processing. Specifically, we use their final per-point predictions to establish point-wise correspondences, and then estimate a single global rigid transformation using standard SVD-based alignment. The results are shown in Table 2. Our method achieves strong performance across all four rigid benchmarks: ModelNet, 3DMatch, 3DLoMatch, and KITTI. On ModelNet, we obtain an RRE of 1.452 and RTE of 0.013, outperforming MAC, FastMAC, and GeoTransformer. We also achieve the lowest RRE and RTE on both 3DMatch and the more challenging low-overlap 3DLoMatch, demonstrating robustness in real indoor scenes. On KITTI, our method yields an RRE of 0.21 and RTE of 0.043, surpassing existing approaches. These results confirm that our framework generalizes well from synthetic to complex real-world indoor and outdoor environments. Qualitative results are provided in Appendix B.

*Table 3.* Comparisons with state-of-the-art methods on CAPE. Bold numbers indicate the best performance, while underlined numbers denote the second best.

| | CAPE | | | |
|---|---|---|---|---|
| | EPE↓ | AccS↑ | AccR↑ | OR↓ |
| PointPWC-Net | 0.039 | 17.9 | 35.7 | 85.9 |
| FLOT | 0.049 | 21.2 | 31.1 | 92.2 |
| DGFM | 0.036 | 35.5 | 62.4 | 73.7 |
| SyNoRiM | 0.030 | 55.5 | 89.1 | 59.1 |
| Lepard+NICP | 0.089 | 23.9 | 44.7 | 78.0 |
| LNDP | 0.045 | 61.3 | 92.2 | 45.7 |
| Geotransformer | 0.055 | 20.9 | 50.1 | 75.5 |
| GraphSCNet | 0.059 | 66.7 | 79.4 | 62.1 |
| Ours | **0.023** | **77.7** | **93.2** | **36.4** |

### 4.4. Analyses on the Generalization

**Dataset.** To evaluate the generalization capability of our method, we test on the CAPE (Ma et al., 2020; Pons-Moll et al., 2017) without any fine-tuning or training on it. CAPE contains the complete scans of dynamic clothed humans.

It consists of 15 human subjects and provides accurate 3D mesh registrations. We use the data preprocessed by (Huang et al., 2023), where each point could contain 8192 points. We note that the CAPE split used in the original Graph-SCNet (Qin et al., 2023) paper contains 11,288 sampled point-cloud pairs, but this split is not publicly available. For reproducibility, we evaluate all methods on the public CAPE test set containing 2,508 point-cloud pairs. Therefore, the reported GraphSCNet result in Table 3 is obtained under our reproducible public test setting and may differ from the number reported in the original paper.

**Quantitative comparisons.** We investigate the generality of our method on CAPE. As shown in Table 3, SyNoRiM-self (Huang et al., 2023) is a self-supervised approach trained directly on CAPE, while GeoTransformer (Qin et al., 2022) is trained on a rigid registration dataset, ModelNet, and evaluated on CAPE in a cross-task setting. GraphSC-Net (Qin et al., 2023), in turn, is trained on 4DMatch and tested on CAPE to assess its non-rigid generalization ability. In contrast, our RGGT is jointly trained on both rigid (ModelNet) and non-rigid (4DMatch) datasets, aiming to learn a unified representation capable of transferring across deformation types.

Across all metrics, our method achieves a clear performance margin over the baselines. RGGT reduces EPE to 0.023 and significantly boosts AccS and AccR to 77.7 and 93.2, respectively, outperforming SyNoRiM-self even though SyNoRiM-self is trained on the same dataset. Compared with GeoTransformer and GraphSCNet, both of which struggle due to domain and deformation gaps, our model yields substantially lower OR and exhibits much stronger correspondence reliability. These results demonstrate that RGGT generalizes more effectively across datasets and deformation regimes, even without CAPE-specific training.

### 4.5. Ablation Studies

We conduct ablation studies to evaluate the effect of the main components in RGGT, as summarized in Table 4. Since Self-Attention (SA) and Cross-Attention (CA) are standard components widely used in transformer-based registration frameworks, we ablate only the newly introduced Global Attention (GA) module. Removing GA leads to a clear drop in AccS and AccR, indicating that incorporating global contextual information helps improve the overall alignment quality. We also observe that adding the Correspondence Loss (CL) further improves performance, particularly on accuracy-based metrics. Finally, incorporating Trellis Features (TF) provides the largest overall gain across all metrics, showing that generative-prior features strengthen the learned representations. The full model, combining all components, achieves the best performance on all evaluation metrics, demonstrating that these components complement

each other in our unified registration framework.

*Table 4.* Ablation experiments. Bold numbers indicate the best performance.

| RL | CL | GA | TF | EPE↓ | AccS↑ | AccR↑ | OR↓ |
|----|----|----|----|------|-------|-------|-----|
| √ | | | | 0.047 | 57.6 | 75.5 | 13.5 |
| √ | √ | | | 0.045 | 59.2 | 77.7 | 12.3 |
| √ | √ | √ | | 0.039 | 62.1 | 79.6 | 11.3 |
| √ | √ | √ | √ | **0.031** | **73.9** | **89.7** | **7.1** |

### 4.6. Why TRELLIS over PartField?

Considering PartField (Liu et al., 2025) as a large-scale representation learning model with demonstrated feature transferability across diverse shapes (Zhu et al., 2025), we additionally adopt it as an alternative 3D prior feature extractor to investigate how different feature representations affect registration performance. Specifically, we replace the original TRELLIS module with PartField while keeping all other components unchanged, and evaluate the resulting model on the challenging 4DMatch benchmark.

Quantitative results show that both TRELLIS and PartField provide strong 3D priors, with TRELLIS showing a slight advantage in our current setting. In particular, TRELLIS achieves a lower EPE (0.031 vs. 0.032), higher AccS (73.9 vs. 72.2), higher AccR (89.7 vs. 88.8), and a lower outlier ratio OR (7.1 vs. 7.4). One possible explanation is that TRELLIS, as a 3D generative model, may provide denser geometry-aware cues that are favorable for point-level correspondence. PartField, on the other hand, provides reliable part-level structural features, which can also benefit registration but may be less fine-grained for dense point-wise matching in this setting. Based on these empirical results, we adopt TRELLIS as the default feature encoder in our framework.

## 5. Conclusion

In this paper, we presented RGGT, a Generative-Prior-Guided Transformer that achieves unified and robust performance on both rigid and non-rigid point cloud registration. By integrating geometric-semantic feature priors from TRELLIS with a Global-Self-Cross Attention architecture and a dual-level supervision strategy, RGGT effectively bridges the representational and optimization gaps between the two registration paradigms. Extensive experiments on rigid, non-rigid, and cross-dataset generalization benchmarks demonstrate that our model not only achieves state-of-the-art accuracy and robustness but also exhibits strong cross-task generalization.

## Acknowledgements

This work was supported by the National Natural Science Foundation of China (No. T2322012, No. 62572240); the Faculty Research Grants (SDS24A8, SDS25A15 and SDS24A19), the Interdisciplinary & Strategic Research Grant (ISRG252606), and the Direct Grants (DR25E8 and DR26F2) of Lingnan University, Hong Kong; and a grant from the Research Grants Council of the Hong Kong Special Administrative Region, China (No. UGC/FDS16/E03/24) and the Hong Kong Metropolitan University Research Grant (No. RD/2025/1.32).

## Impact Statement

This paper presents work whose goal is to advance the field of Machine Learning. There are many potential societal consequences of our work, none which we feel must be specifically highlighted here.

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

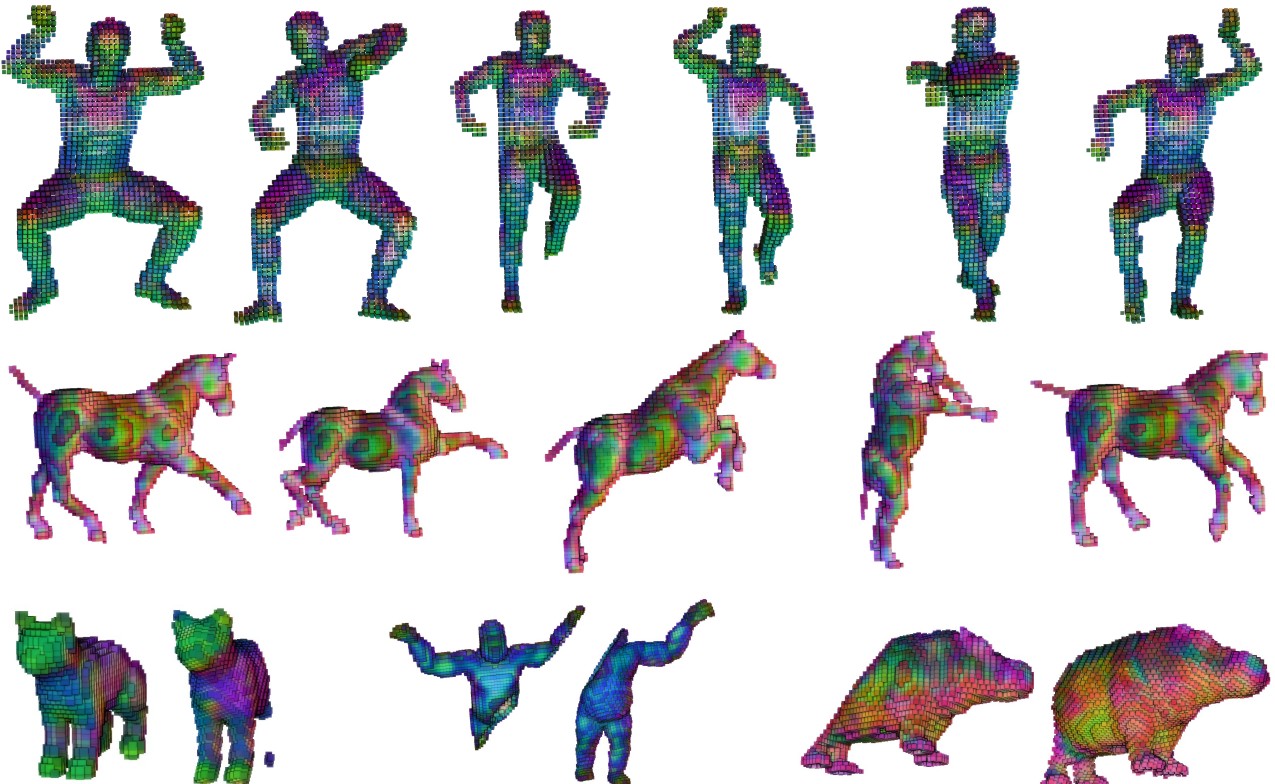

*Figure 4.* More visualization results of TRELLIS features. Top row: non-rigid dancer sequences. Middle row: non-rigid horse sequences. Bottom row: rigid objects, cat, gorilla, and hippo. Color represents the first three components of PCA-computed TRELLIS features, where similar colors indicate semantically and geometrically consistent regions.

## A. More visualization results of TRELLIS features

To better illustrate the structural priors provided by TRELLIS features, we visualize latent features extracted from the Sparse Flow Transformer across a wide range of deformation patterns and object categories, as shown in Figure 4. The first row shows multiple dancer poses from the non-rigid dataset, while the second row presents several horse examples. To highlight the generality of TRELLIS beyond non-rigid motion, the last row includes three representative rigid shapes (cat, gorilla and hippo). Across all categories, TRELLIS produces semantically consistent multi-scale feature distributions, where corresponding regions share similar color patterns despite large pose, shape, or structural variations. These visualizations demonstrate that TRELLIS encodes local geometric structure and global semantic coherence that generalize well across both rigid and non-rigid objects.

## B. Qualitative rigid registration results

In Figure 5, we present qualitative results of rigid registration on five settings: two object categories from ModelNet (Airplane and Car), as well as the real-world datasets 3DMatch, 3DLoMatch, and KITTI. Each column corresponds to a distinct category or scene type, demonstrating the generalization of our approach across synthetic shapes and real-world indoor/outdoor environments. The comparison between the ground-truth alignment and our predicted registration in the last two rows shows that our method produces accurate and well-aligned transformations. These examples further confirm the robustness and effectiveness of RGGT on rigid registration tasks.

## C. Qualitative non-rigid registration results on the 4DMatch dataset

As shown in Figure 6, we visualize the predicted alignments and corresponding error maps across different methods. On 4DMatch, RGGT achieves the most accurate alignment, generating smooth and precise deformation fields with minimal residual errors. In the *deer* case (top row), LNDP struggles to capture large local motions, leading to misalignments in

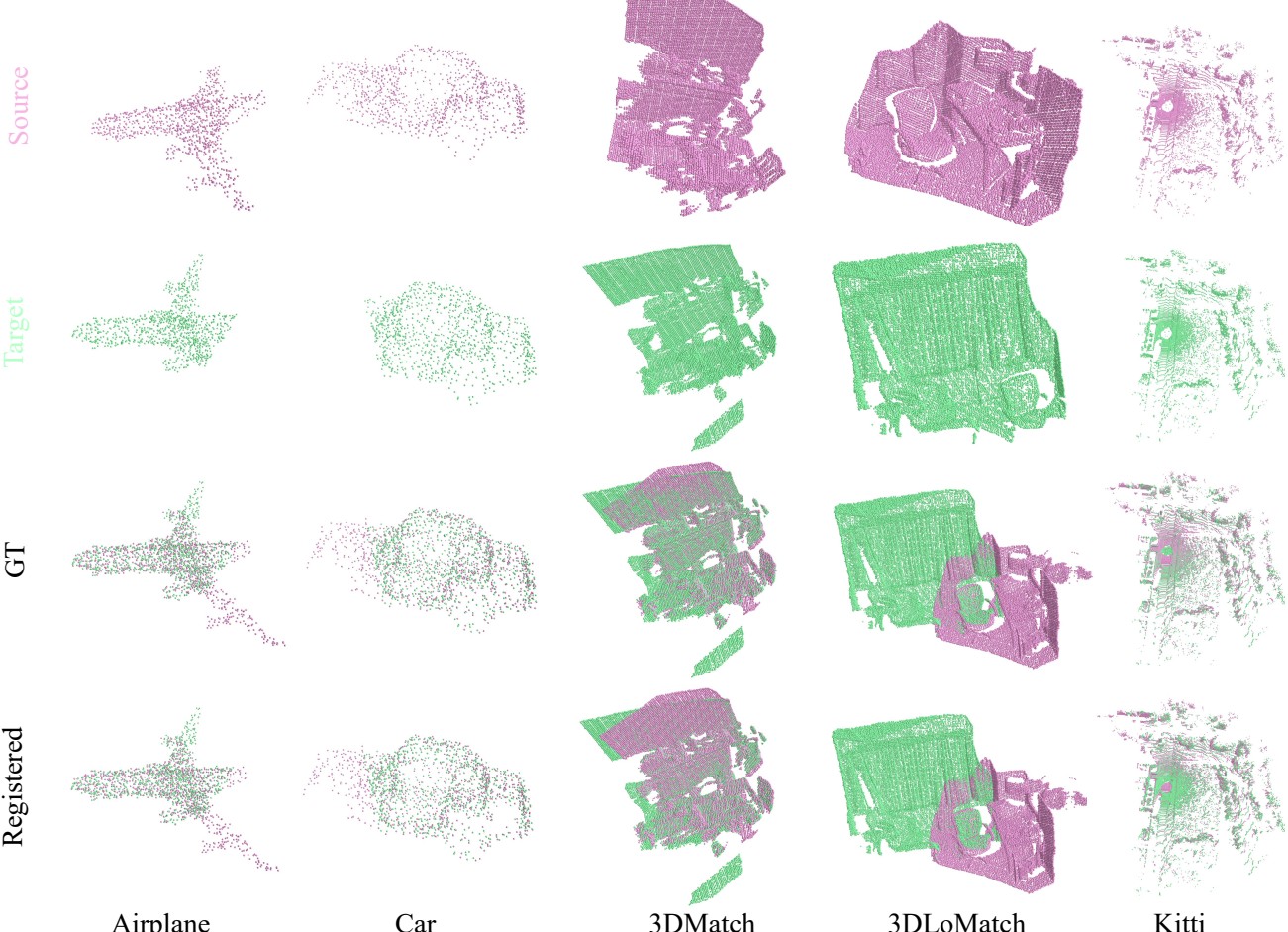

*Figure 5.* Qualitative rigid registration results across multiple datasets. From top to bottom: source point clouds, target point clouds, ground-truth alignment, and the registration results produced by our method. From left to right: examples from ModelNet (Airplane, Car), 3DMatch, 3DLoMatch, and KITTI.

the hind legs and antlers. GraphSCNet maintains local rigidity but underfits complex articulations, resulting in residual misalignments around the limbs and ears, as shown in the rabbit example (middle row). In contrast, our method achieves coherent, structure-preserving correspondences in both *deer* and *rabbit* examples, maintaining global consistency and local geometric fidelity. On 4DLoMatch, as shown in the bottom row of Figure 6, even under low-overlap and partially occluded conditions such as the *dancer* example, we still produce smooth and continuous alignments, while competing methods exhibit severe distortions or fragmented results.

## D. Qualitative generalization results on the CAPE dataset

We further provide qualitative generalization registration results on the CAPE dataset, as shown in Fig. 7. Each example is visualized across two successive rows. The leftmost column shows the source and target point clouds, while the remaining columns present the outputs of GeoTransformer, GraphSCNet, SyNoRiM, and our method. For each method, the top row displays the predicted alignment, and the bottom row provides the corresponding per-point error map, allowing a clear comparison of alignment accuracy across approaches. Across all examples, GeoTransformer exhibits noticeable misalignments in articulated regions such as the arms and hands, highlighting its limited ability to handle large pose variations under rigid assumptions. GraphSCNet alleviates some local inconsistencies but still produces residual errors around the limbs, often resulting in slight shifts between the predicted and ground-truth alignments. SyNoRiM performs better on CAPE due to its self-supervised training on the same dataset, yet it still struggles when the subject undergoes extreme arm or torso movements. In contrast, RGGT generates consistently accurate rigid alignments across diverse

poses. Our method sharply reduces errors at the extremities and maintains overall body consistency, demonstrating strong cross-dataset generalization despite never being trained on CAPE.

## E. Transformation and Deformation Estimation

Our framework supports both rigid registration and non-rigid deformation in a unified manner. Given the point-wise output from our network, where each source keypoints $p_i' \in P'$ is directly associated with its predicted target position $\tilde{p}_i' \in \tilde{P}'$, we leverage this implicit correspondence to estimate either a global rigid transformation or a smooth non-rigid deformation field, depending on the nature of the input data.

For rigid scenarios (e.g., ModelNet40, 3DMatch, KITTI), we compute a single rigid transformation $(\mathbf{R}, \mathbf{t})$ that best aligns $P'$ to $\tilde{P}'$ by solving:

$$\mathbf{R}, \mathbf{t} = \arg \min_{\mathbf{R} \in \mathrm{SO}(3), \mathbf{t} \in \mathbb{R}^3} \sum_{i \in P'} \|\mathbf{R}p_i' + \mathbf{t} - \tilde{p}_i'\|_2^2, \tag{10}$$

which admits a closed-form solution via SVD (Besl & McKay, 1992).

For non-rigid cases (e.g., 4DMatch, CAPE), we instead model the motion as a spatially varying deformation. We construct an embedded deformation graph $\hat{\mathcal{G}} = \{\hat{\mathcal{V}}, \hat{\mathcal{E}}\}$ over $P$, where each node $\hat{\mathbf{v}}_j$ is assigned a local rigid transformation $(\mathbf{R}_j, \mathbf{t}_j)$. The deformed position of any point $p_i$ is then given by:

$$\mathcal{W}(p_i) = \sum_{j \in \mathcal{N}_i} \alpha_{i,j} \left( \mathbf{R}_j (p_i - \hat{\mathbf{v}}_j) + \mathbf{t}_j + \hat{\mathbf{v}}_j \right), \tag{11}$$

where $\alpha_{i,j}$ is the skinning factor as in DynamicFusion (Newcombe et al., 2015):

$$\alpha_{i,j} = \frac{\exp(-\|p_i - \mathbf{v}_j\|^2 / (2\sigma_n^2))}{\sum_{k \in \mathcal{N}_i} \exp(-\|p_i - \mathbf{v}_k\|^2 / (2\sigma_n^2))}. \tag{12}$$

Each local transformation $(\mathbf{R}_j, \mathbf{t}_j)$ is computed independently by solving a weighted rigid alignment problem over the neighborhood of node $\hat{\mathbf{v}}_j$:

$$\mathbf{R}_j, \mathbf{t}_j = \arg \min_{\mathbf{R} \in \mathrm{SO}(3), \mathbf{t} \in \mathbb{R}^3} \sum_{i \in P' \cap \mathcal{N}_j} w_{i,j} \|\mathbf{R}p_i' + \mathbf{t} - \tilde{p}_i'\|_2^2, \tag{13}$$

where $\mathcal{N}_j$ denotes the set of points influenced by node $j$, and the weight $w_{i,j}$ is typically chosen as the skinning factor $\alpha_{i,j}$. This problem admits a closed-form solution via weighted SVD (Besl & McKay, 1992), enabling efficient and parallel computation of all local transformations.

Thus, the same network prediction $\{\tilde{p}_i'\}$ serves as the foundation for both estimation modes, enabling a flexible and task-adaptive registration pipeline.

## F. Running Efficiency

We evaluate runtime on the 4DMatch dataset. Extracting TRELLIS features requires around 10 s per pair (one-time preprocessing), while RGGT inference runs in 0.325 s. For comparison, LNDP requires 2.5 s, GeoTransformer 0.22 s, and GraphSCNet 0.22 s + 0.301 s for its two-stage process. Although TRELLIS introduces additional preprocessing cost, RGGT achieves competitive inference speed once features are available, offering an effective balance between accuracy and efficiency.

## G. Other Ablation Studies

**Effect of Text Conditioning.** To examine whether RGGT critically relies on explicit category text, we replace the category prompt in TRELLIS with an empty prompt on ModelNet. As shown in Table 5, RGGT with an empty prompt achieves 1.466 / 0.017 RRE/RTE, which is close to the full model using category prompts, 1.452 / 0.013. This suggests that the dominant signal comes from TRELLIS's native 3D structural prior, while text conditioning mainly acts as a mild semantic regularizer.

**Training Fairness Analysis.** Since RGGT is trained in a mixed manner across multiple datasets, we further conduct a controlled analysis on ModelNet to examine whether the improvement mainly comes from multi-dataset joint training. As shown in Table 6, retraining GeoTransformer on all four datasets only slightly improves its performance from 2.564 / 0.026 to 2.512 / 0.022 in terms of RRE/RTE. In contrast, RGGT trained only on ModelNet still achieves 1.513 / 0.015, which remains clearly better than GeoTransformer under both training settings. This suggests that the performance gain is not solely caused by multi-dataset joint training.

Table 5. Effect of text conditioning on ModelNet.

| Setting/Method | RRE ↓ | RTE ↓ |
| --- | --- | --- |
| RGGT w/ category prompt | 1.452 | 0.013 |
| RGGT w/ empty prompt | 1.466 | 0.017 |

Table 6. Training fairness analysis on ModelNet.

| Method | Training data | RRE ↓ | RTE ↓ |
| --- | --- | --- | --- |
| GeoTransformer | ModelNet only | 2.564 | 0.026 |
| GeoTransformer | Four datasets | 2.512 | 0.022 |
| RGGT | ModelNet only | 1.513 | 0.015 |
| RGGT | Four datasets | 1.452 | 0.013 |

Table 7. Controlled analysis of TRELLIS features and RGGT backbone on ModelNet.

| Setting (only trained on ModelNet ) | RRE ↓ | RTE ↓ |
| --- | --- | --- |
| GeoTransformer | 2.564 | 0.026 |
| GeoTransformer + TRELLIS features | 2.131 | 0.022 |
| RGGT w/ GeoTransformer backbone | 1.723 | 0.019 |
| Full RGGT | 1.513 | 0.015 |

**Effect of TRELLIS Features and RGGT Backbone.** To further isolate the effects of TRELLIS features and the proposed RGGT backbone, we conduct two controlled experiments on ModelNet. First, we plug TRELLIS features into GeoTransformer. As shown in Table 7, this improves GeoTransformer from 2.564 / 0.026 to 2.131 / 0.022 in terms of RRE/RTE, indicating that TRELLIS features provide useful registration-relevant priors. However, this result is still worse than RGGT. Second, we replace the RGGT backbone with the GeoTransformer module while keeping the remaining components unchanged, which obtains 1.723 / 0.019. This is also inferior to the full RGGT. These results suggest that the final gain comes from the combination of TRELLIS generative priors and the proposed RGGT backbone, rather than TRELLIS features alone.

**Sensitivity to TRELLIS Backbone Scale.** We further provide a preliminary inference-stage sensitivity analysis of the TRELLIS backbone scale on ModelNet. In this analysis, we replace the TRELLIS checkpoint while keeping the remaining RGGT model fixed, instead of retraining RGGT from scratch for each scale. As shown in Table 8, larger TRELLIS backbones lead to better performance, where the 342M, 1.1B, and 2B checkpoints achieve 1.974 / 0.021, 1.764 / 0.018, and 1.452 / 0.013 RRE/RTE, respectively. Meanwhile, RGGT remains effective with smaller checkpoints, suggesting that the method is not tied to a specific 2B checkpoint, although backbone scale does affect the final accuracy.

**Quantitative Analysis of TRELLIS Features.** Besides the qualitative PCA visualization, we further quantify whether TRELLIS features carry registration-relevant information by performing nearest-neighbor matching directly in the TRELLIS feature space. On ModelNet, this simple matching strategy achieves 57.5 IR and 98.5 FMR, which is comparable to FCGF (Choy et al., 2019) with 56.8 IR and 97.4 FMR. This confirms that TRELLIS features already encode useful correspondence information before being processed by the RGGT backbone.

*Table 8.* Preliminary inference-stage sensitivity analysis of TRELLIS backbone scale on ModelNet.

| TRELLIS checkpoint | RRE ↓ | RTE ↓ |
|---|---|---|
| 342M | 1.974 | 0.021 |
| 1.1B | 1.764 | 0.018 |
| 2B | 1.452 | 0.013 |

*Table 9.* Quantitative analysis of TRELLIS features on ModelNet.

| Feature / Method | IR ↑ | FMR ↑ |
|---|---|---|
| FCGF (Choy et al., 2019) | 56.8 | 97.4 |
| TRELLIS NN matching | 57.5 | 98.5 |

## H. Additional Comparisons

**Comparison with DV-Matcher, SHOT, and HKS.** We additionally compare RGGT with DV-Matcher, SHOT, and HKS on ModelNet under the same evaluation setting. As shown in Table 10, DV-Matcher achieves 2.457 / 0.022 RRE/RTE, while the classical descriptors SHOT and HKS obtain 16.53 / 0.163 and 25.32 / 0.160, respectively. RGGT achieves lower registration errors than these additional baselines, further validating the effectiveness of the proposed generative-prior-guided representation.

## I. Failure Case Analysis

Although RGGT achieves strong performance on established low-overlap benchmarks, it can still fail under extremely low-overlap conditions. Figure 8 shows a representative failure case on ModelNet, where the overlap ratio between the source and target point clouds is only 3.58%. In this case, the shared geometric evidence is too limited to support reliable correspondence estimation, leading to an inaccurate alignment. This suggests that, despite the benefit of generative-prior-guided features, extremely low overlap remains a challenging scenario for our method.

## J. Limitations and Future Work

Although KPConv downsampling reduces the attention cost by applying the transformer to compact keypoints rather than full-resolution point clouds, scaling RGGT to extremely large-scale scenes remains challenging, and future work may incorporate sparse or efficient attention mechanisms. In addition, RGGT assumes a valid shared correspondence structure in the overlapping region. Severe topology changes, such as parts appearing, disappearing, or changing connectivity within the overlap, may violate this assumption and cause unreliable matching. Explicitly handling topology changes remains an important future direction.

*Table 10.* Additional comparisons on ModelNet.

| Method | RRE ↓ | RTE ↓ |
|---|---|---|
| SHOT (Salti et al., 2014) | 16.53 | 0.163 |
| HKS (Sun et al., 2009) | 25.32 | 0.160 |
| DV-Matcher (Chen et al., 2025) | 2.457 | 0.022 |
| RGGT | 1.452 | 0.013 |

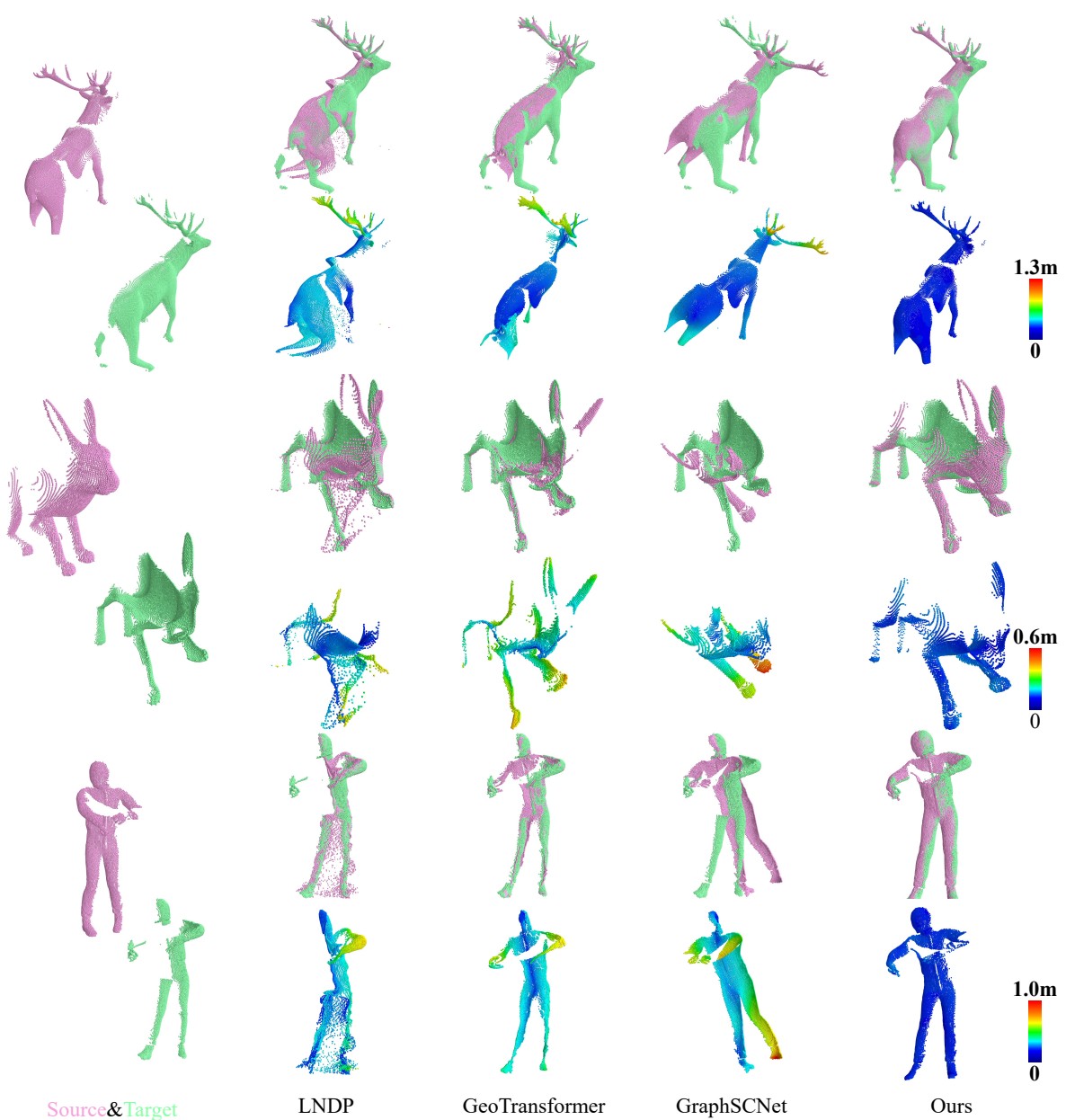

Source&Target     LNDP     GeoTransformer     GraphSCNet     Ours

*Figure 6.* Registration visualization results on 4DMatch and 4DLoMatch. For each case, we display the point cloud alignment (top) and its corresponding per-point error map (bottom). The top two rows show non-rigid examples from 4DMatch, while the bottom row illustrates a challenging low-overlap case from 4DLoMatch. Compared with LNDP, GeoTransformer, and GraphSCNet, RGGT produces smoother and more coherent deformations with significantly fewer misaligned regions.

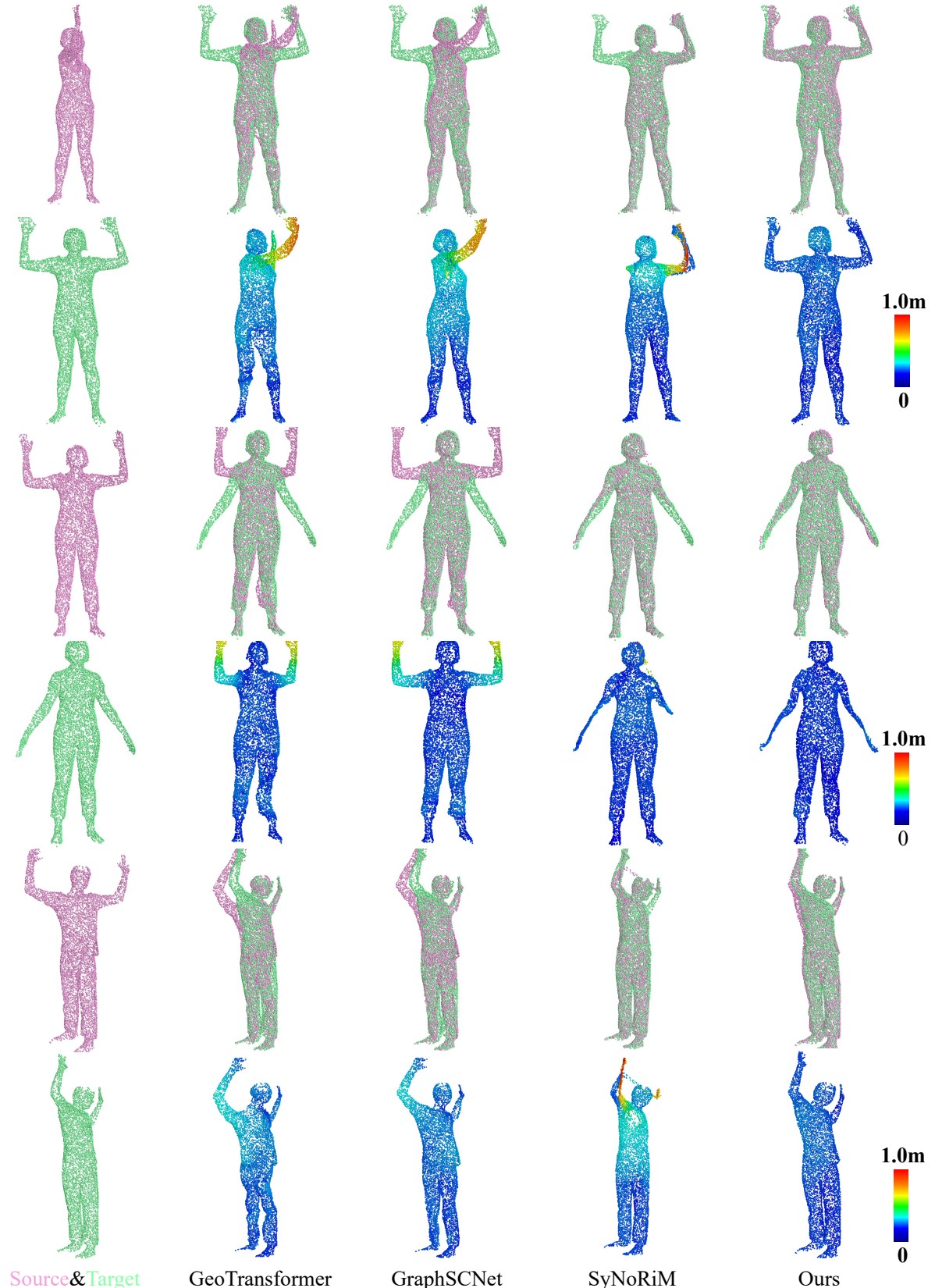

Source&Target    GeoTransformer    GraphSCNet    SyNoRiM    Ours

*Figure 7.* Qualitative non-rigid registration results on the CAPE dataset.

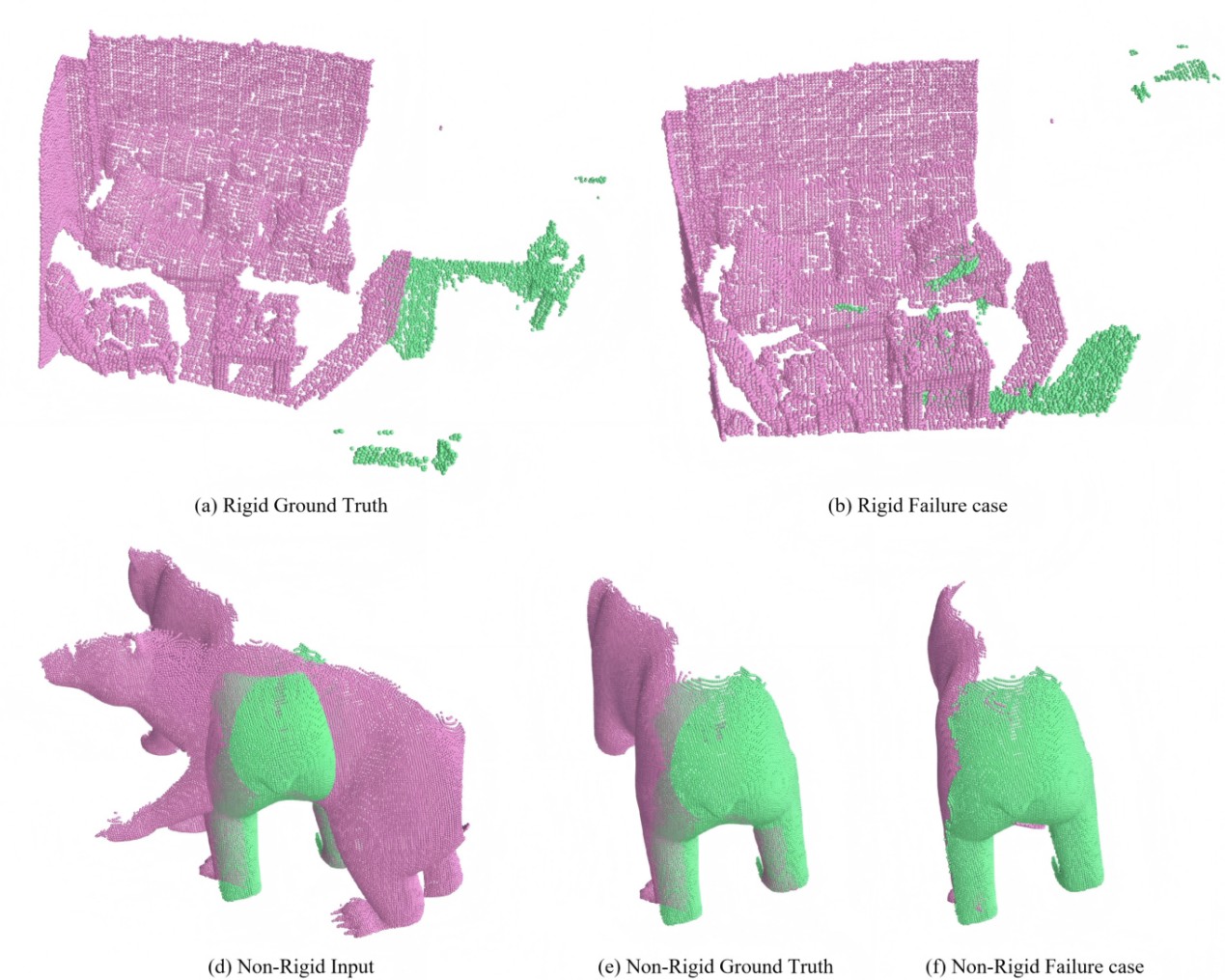

(a) Rigid Ground Truth

(b) Rigid Failure case

(d) Non-Rigid Input

(e) Non-Rigid Ground Truth

(f) Non-Rigid Failure case

*Figure 8.* Figure 1: Failure cases under low overlap. The top row shows a rigid registration failure under extremely low overlap(3.58%). The very small shared region provides insufficient reference information from the target point cloud, leading to failure in predicting the transformed source coordinates. The bottom row shows a non-rigid registration failure under low overlap (14.2%). Although the predicted shape is roughly aligned with the target, substantial local discrepancies remain, indicating that low overlap still limits fine-grained deformation estimation.

