# OpenReview forum: "RGGT: A Generative-Prior-Guided Transformer for Unified Rigid and Non-Rigid Point Cloud Registration"
_ICML.cc/2026/Conference — ICML 2026 regular_

### Official Review · Reviewer_x2Ab · 2026-02-15

**Soundness:** 3
**Presentation:** 3
**Significance:** 3
**Originality:** 3
**Overall Recommendation:** 4
**Confidence:** 4

**Summary:**

This paper proposes the an unified framework for both rigid and non-rigid point cloud registration within a shared optimization space. The key insight is that rigid registration relies on stable cross-scale geometric similarity while non-rigid registration requires locally coherent but globally semantically driven correspondences—a multi-scale discrepancy that creates an optimization gap. RGGT bridges this gap through three coordinated designs: (1) generative priors distilled from TRELLIS (a native 3D diffusion model) enrich point features with unified geometric-semantic cues; (2) a Global-Self-Cross Attention module jointly models long-range structure, local interaction, and bidirectional cross-shape reasoning; and (3) a dual correspondence-reconstruction objective provides consistent supervision.

**Compliance With Llm Reviewing Policy:**

Affirmed.

**Final Justification:**

This paper proposes the an unified framework for both rigid and non-rigid point cloud registration within a shared optimization space. I hope the author can add the rebuttal additional results into the future version, and I consider weak accept.

**Key Questions For Authors:**

1. Could the TRELLIS feature extraction be accelerated or made end-to-end trainable to eliminate the preprocessing bottleneck?
2. How would RGGT handle scenarios with topology changes (e.g., objects appearing/disappearing) that violate the correspondence assumption?
3. What architectural modifications would be necessary to truly unify the transformation estimation rather than requiring separate post-processing steps?
4. How sensitive is performance to the quality of TRELLIS features? Would the framework still work with less powerful 3D generative priors?

**Limitations:**

No. Some limitations of the work could be included.

**Strengths And Weaknesses:**

## Strengths
1. A framework to successfully unify rigid and non-rigid registration with competitive performance on both tasks.
2. Innovative use of native 3D generative model (TRELLIS) features that preserve both fine-grained geometry and global semantics.
3. Comprehensive evaluation across diverse benchmarks demonstrating strong cross-task generalization (e.g., on CAPE without fine-tuning).
4. Clear ablation studies validating the contribution of each component (global attention, correspondence loss, TRELLIS features).

## Weaknesses
1. Some related works comparisons are missed and there is less of discussion, such as DV-Matcher [1], which also uses attention-based network. Some other axiomatic-based method could also be included, such as SHOT [2], HKS [3], ect.
TRELLIS feature extraction requires significant preprocessing time, creating a practical bottleneck despite fast inference.
2. Limited analysis of failure cases in extremely challenging scenarios (e.g., severe occlusions, topology changes).
3. The claim of "unified" registration might be somewhat overstated since the framework still requires post-hoc estimation of either rigid transformation or non-rigid deformation field based on task type.
4. Computational complexity of the three-attention mechanism may limit scalability to very large point clouds.

[1] DV-Matcher: Deformation-based Non-Rigid Point Cloud Matching Guided by Pre-trained Visual Features.
[2] Shot: unique signatures of histograms for surface and texture description.
[3] A concise and provably informative multi-scale signature based on heat diffusion.

---

> ### Author Rebuttal · Authors · 2026-03-30
>
> We thank the reviewer for the suggestion.
> ### Issue 1: Related Work
> We will add discussion of DV-Matcher, SHOT, and HKS, and clarify their differences from our work.
> We additionally evaluated on ModelNet: DV-Matcher, SHOT, and HKS obtain RRE/RTE of 2.457/0.022, 16.53/0.163, and 25.32/0.160, respectively.
> These results provide a more direct empirical comparison with our method.
> ### Issue 2: Failure Cases
> Under severe occlusion or extremely low overlap, RGGT may still fail. For instance, with only 3.58% overlap, the shared geometric evidence is too limited to support reliable alignment. We provide a representative failure case via the anonymous link in reviewer nujG Issue 1. For topology changes, we will provide a more explicit analysis in the response/discussion for Issue 6.
> ### Issue 3: "Unified" Claim
> We agree that “unified” could be ambiguous if interpreted as requiring identical post-processing for rigid and non-rigid settings.
> Here, “unified” refers to a shared backbone, representation, and training objective across both tasks, not a shared final estimator.
> The task-dependent part appears only in the final post-processing step. To avoid ambiguity, we will tone down the wording.
> ### Issue 4: Scalability
> We agree that the computational cost of the three-attention mechanism increases with the number of input points, which may limit scalability if applied directly to very large point clouds.
> In practice, we apply the transformer after KPConv downsampling, so attention operates on keypoints rather than the full-resolution cloud.
> This keeps computation manageable, although improving scalability to larger scenes remains an important direction for future work, for example by incorporating more efficient attention designs such as log-linear sparse attention.
> ### Issue 5: Preprocessing Bottleneck
> We agree that Trellis extraction is a preprocessing bottleneck, although RGGT itself is efficient once features are available.
> Because Trellis is diffusion-based, it can benefit from recent acceleration methods [a-c], making preprocessing acceleration feasible.
> A more tightly integrated or end-to-end trainable pipeline is another possible direction to reduce this overhead.
>
> [a] Improving the training of rectified flows.
>
> [b] PeRFlow: piecewise rectified flow as universal plug-and-play accelerator.
>
> [c] Trainable Log-linear Sparse Attention for Efficient Diffusion Transformers.
> ### Issue 6: Topology Changes
> Firstly, we understand that in non-rigid registration, the overlapping region is allowed to deform, sometimes substantially, as long as a valid shared correspondence structure still exists between source and target.
> By contrast, topology changes (e.g., parts appearing or disappearing) may break this assumption because one-to-one correspondences may no longer exist in the shared region.
> Our current framework does not explicitly handle such topology changes.
> If they occur within the overlap, the method may fail.
> If they occur outside the overlapping region, their impact is usually limited.
> We will clarify this assumption in the revision.
> ### Issue 7: Truly Unify the Transformation Estimation
> We agree that a truly unified formulation would need to unify not only the shared backbone and supervision, but also the final transformation parameterization.
>
> In our design, the main reason for task-dependent post-processing is that the network predicts correspondences/reconstructed positions only on a downsampled keypoint set.
> This requires a separate geometric recovery step: a global SVD solver for rigid registration, and a deformation-field estimator for non-rigid registration.
> A more fully unified estimator would therefore need full-resolution dense prediction.
>
> More fundamentally, it would require a single transformation family that subsumes both rigid and non-rigid motion.
> One promising direction is to predict a dense motion field directly, which would in practice already solve both non-rigid and rigid registration.
> Our current work does not yet include this fully unified estimator, and instead unifies the tasks at the levels of representation, backbone, and training objective. We will clarify this distinction in the revision.
> ### Issue 8: Feature Sensitivity
> We agree that the framework is sensitive to the quality of the input prior features: weaker correspondence cues generally lead to worse registration performance.
> This is supported by our experiments: removing Trellis causes a clear drop, while replacing it with PartField remains competitive but slightly inferior.
> These results suggest that the framework itself is not specific to Trellis, but stronger 3D priors lead to better performance.
>
> In other words, RGGT can in principle work with less powerful 3D priors, but its performance depends on the quality of the correspondence-relevant information they provide. The best results come from the combination of a strong 3D prior, the RGGT architecture, and the joint training objective. We will clarify this point in the revision.

---

> > ### Author Rebuttal · Reviewer_x2Ab · 2026-04-01
> >
> > Thanks for the author's reply. I hope the author can add these additional results into the future version, and I remain my original score.

---

> > > ### Author Response · Authors · 2026-04-04
> > >
> > > Thank you very much for your thoughtful follow-up. We sincerely appreciate your helpful feedback and constructive suggestions. We agree that the additional results you suggested would further strengthen the paper, and we will include them in the future version. If any point would benefit from further clarification, we would be happy to provide more details.

---

### Official Review · Reviewer_nyiA · 2026-03-10

**Soundness:** 2
**Presentation:** 3
**Significance:** 2
**Originality:** 2
**Overall Recommendation:** 4
**Confidence:** 4

**Summary:**

A central concept outlined by this paper is the unification of rigid and non-rigid point cloud registration within a shared optimization space. This submission's key finding concerns the effectiveness of leveraging 3D generative priors (specifically from TRELLIS) to provide geometric-semantic cues that bridge the representational gap between rigid consistency and non-rigid deformation modeling. The paper proposes RGGT, which integrates TRELLIS features with a Global-Self-Cross Attention transformer and a dual correspondence-reconstruction objective. The method achieves state-of-the-art results on both rigid (ModelNet40, 3DMatch, KITTI) and non-rigid (4DMatch) benchmarks within a single unified framework.

**Compliance With Llm Reviewing Policy:**

Affirmed.

**Key Questions For Authors:**

1. How does the method handle cases where the shape name (text prompt for TRELLIS) is unknown?
2. What is the inference time compared to baselines like GeoTransformer and GraphSCNet?
3. Can the method handle partial point clouds with very low overlap (<30%)?
4. Will code and pretrained models be released?

**Limitations:**

yes

**Strengths And Weaknesses:**

1. Soundness

Strengths:
- The problem formulation (Eq. 1) clearly articulates the unified registration objective
- The TRELLIS feature extraction (Eq. 2) is well-motivated and described
- The attention mechanism design (Eq. 3-5) is principled and comprehensive
- The dual loss formulation (Eq. 6-9) balances feature-level and coordinate-level supervision
- Cross-dataset generalization test on CAPE (Table 3) demonstrates robustness
- Comparison with TRELLIS vs. PartField (Section 4.6) validates the choice of generative prior

Weaknesses:
- The text prompt (shape name) required for TRELLIS feature extraction may not be available in real-world scenarios
- The PCA visualization (Figure 3) shows feature coherence but doesn't quantify it
- The symmetric parameterization of Wf is mentioned but not justified
- No runtime comparison with baselines
- No analysis of failure cases or limitations
- The GeoTransformer baseline on 4DMatch (Table 1) is trained specifically for this comparison, which may not be a fair comparison

2. Presentation

Strengths:
- The paper is well-organized and clearly written
- Figure 1 provides compelling visual evidence of the method's effectiveness
- Figure 2 clearly illustrates the architecture
- The motivation for unifying rigid and non-rigid registration is well-articulated

Weaknesses:
- The distinction between "geometric-semantic" cues could be clarified
- Some implementation details (e.g., how shape names are obtained for TRELLIS) are not explained


3. Originality and Significance

Strengths:
- First framework to unify rigid and non-rigid registration within a shared optimization space
- Novel use of native 3D generative priors (TRELLIS) for registration, avoiding the domain gap of 2D foundation models
- The Global-Self-Cross Attention design effectively captures global context, local geometry, and cross-shape interactions
- The dual correspondence-reconstruction objective provides consistent supervision across deformation types

Weaknesses:
- The use of pre-trained features for registration is not new; the novelty lies in the specific choice of TRELLIS and the unified framework
- The transformer architecture follows established designs (GeoTransformer, REGTR)

---

> ### Author Rebuttal · Authors · 2026-03-30
>
> Thank you for this helpful comment.
> ### Issue1: Trellis Text Conditioning
> We agree that object/category names may be unavailable in real-world scenarios. Using an off-the-shelf 3D classifier or image/point-cloud captioning model to generate approximate prompts is promising for open-world evaluation and unseen categories.
>
> Our additional ablation suggests that RGGT does not critically rely on the text prompt. With an empty prompt on ModelNet, performance remains close (1.466 / 0.017 vs. 1.452 / 0.013 in the original setting). This indicates that the text mainly serves as an auxiliary semantic cue, while the dominant signal comes from the voxelized 3D input and the resulting Trellis prior features.
> ### Issue 2: Runtime Comparison
> A runtime comparison on 4DMatch is included in Appendix F.
> Trellis feature extraction takes about 10 s per pair, while RGGT forward inference takes 0.325s. For comparison, LNDP requires 2.5s, GeoTransformer 0.22s, and GraphSCNet 0.521s in total for its two-stage pipeline (0.22s + 0.301s).
>
> These results show that the main overhead of RGGT comes from Trellis feature extraction rather than the registration backbone itself. This limits real-world efficiency.
> A promising direction is to use faster generative backbones or distill Trellis into a lightweight 3D encoder, retaining generative priors while reducing runtime cost.
> We will clarify this trade-off.
> ### Issue 3: Low-Overlap Robustness
> Our results show that RGGT remains robust on low-overlap benchmarks. In particular, 4DLoMatch contains challenging pairs with overlap below 45%, including cases below 30%, while 3DLoMatch consists entirely of pairs with overlap below 30%. RGGT achieves good performance on both benchmarks, indicating good robustness under limited overlap. That said, RGGT can still fail in *extremely* low-overlap cases, where shared geometric evidence becomes insufficient for reliable registration.
> We have added representative failure-case visualizations and discussion in response to Reviewer nujG, Issue 1.
> ### Issue 4: Code Release
> We will release the code upon acceptance.
> ### Issue 5: Fairness of GeoTransformer Training
> We agree that including GeoTransformer in Table 1 may cause confusion, since it is not designed for non-rigid registration and was included only as an additional cross-task reference related to GraphSCNet, rather than a main baseline. To avoid misunderstanding, we will remove the GeoTransformer result from Table 1 and clarify the comparison protocol.
> ### Issue 6: Symmetric Wf
> In $f(p,c) = \exp(\bar{f}_p^\top W_f \bar{f}_c)$, $\bar{f}_p$ and $\bar{f}_c$ are extracted by the same Trellis encoder and lie in the same feature space. This makes the two inputs symmetric in the similarity computation.
> We therefore constrain $W_f$ to be symmetric, which respects this structure and reduces redundant parameters.
> ### Issue 7: Geometric-semantic Clarification
> We agree that the term "geometric-semantic" needs clarification.
> In our paper, we use this term to describe the type of information reflected in the Trellis features, rather than to imply a strict separation between geometric and semantic components. More specifically, Trellis features are derived from DINOv2 features and can reflect both semantic and geometric information. This usage is motivated by prior work showing that DINOv2 features are useful for both semantic correspondence and geometry-related tasks such as depth estimation. We therefore use the term "geometric-semantic" to indicate that Trellis features may carry both kinds of information useful for registration.
> ### Issue 8: Transformer Design
> Our transformer is related to prior architectures such as GeoTransformer and REGTR, but is not a direct reuse of the standard self-cross design.
> Unlike prior methods that mainly alternate self- and cross-attention, our model adds a global attention branch to aggregate scene-level context before local matching.
> We also provide empirical evidence that this difference is meaningful. As discussed in our response to reviewer e47c Issue 3, when we replace our transformer with GeoTransformer’s while keeping the rest of the pipeline unchanged, the performance on ModelNet drops from 1.513 / 0.015 to 1.723 / 0.019. This suggests that the gain does not come only from Trellis features, but also from the proposed transformer design.
> ### Issue 9: Trellis Feature Quantification
> We agree that Figure 3 is qualitative and does not directly quantify the usefulness of Trellis features. To address this, we add a quantitative analysis on ModelNet by performing nearest-neighbor matching in the Trellis feature space and evaluating the resulting correspondences using IR and FMR. This yields 57.5 IR and 98.5 FMR. Notably, these results are comparable to earlier registration methods. For example, FCGF reports 56.8 IR and 97.4 FMR. This confirms that Trellis features alone already carry registration-relevant information. We will add these results and revise the discussion.

---

> > ### Author Rebuttal · Reviewer_nyiA · 2026-04-01
> >
> > Thanks for the author's reply. I hope the author can make these significant modifications in the future version, and I remain my original score.

---

> > > ### Author Response · Authors · 2026-04-04
> > >
> > > Thank you very much for your thoughtful follow-up. We sincerely appreciate your thoughtful feedback and your constructive suggestions. We fully acknowledge the importance of the significant modifications you pointed out, and we will incorporate them in the future version. If any aspect requires further clarification, we would be happy to elaborate.

---

### Official Review · Reviewer_nujG · 2026-03-11

**Soundness:** 2
**Presentation:** 3
**Significance:** 3
**Originality:** 3
**Overall Recommendation:** 4
**Confidence:** 4

**Summary:**

This paper presents RGGT, a unified network designed for both rigid and non-rigid point cloud registration. Specifically, the method employs the Spatial Feature Transform (SFT) from TRELLIS to extract structural-semantic representations from voxelized point clouds. To reduce computational complexity, KPConv is utilized to downsample the points before they enter the Transformer blocks. The model is optimized using two loss functions reconstruction loss and partial semantic errors to evaluate registration performance. All in all, the paper provides visualization results to facilitate understanding and validates the method's performance and generalizability through comparisons with SOTA rigid and non-rigid registration methods.

**Compliance With Llm Reviewing Policy:**

Affirmed.

**Key Questions For Authors:**

1: TRELLIS is a large generative model originally designed for text/image-to-3D generation. During the feature extraction stage, a text description is provided as input. Could the authors clarify whether this textual prompt targets the object-level semantics or fine-grained geometric structures?

2: Have the authors experimented with using TRELLIS as a point cloud encoder without any textual input? If so, what impact would this have on the registration performance?

**Limitations:**

The paper does not explicitly discuss the limitations of the proposed method. Most of the current bottlenecks stem from the computational performance of TRELLIS, which could potentially be improved by optimizing the Transformer architecture and the 3D implicit representation to enhance runtime efficiency.

**Strengths And Weaknesses:**

Soundness: While the paper presents visually compelling results, the technical validation has notable gaps. Absence of failure case analysis and visualization limits the reader's ability to assess the method's robustness under challenging conditions. Moreover, the PCA analyse of TRELLIS features (e.g., Fig. on page 12 regarding human leg deformations) is not entirely convincing: the correspondence between feature variations and local geometric deformations in partially deformed regions lacks rigorous justification. Strengthening the empirical analysis with ablation studies on feature interpretability and explicit discussion of failure modes would significantly improve the technical soundness of the claims.

Presentation: The paper is clearly written and well-structured. A notable strength lies in the effectiveness of its visualizations: the use of color coding on point clouds and voxels to represent similarity and distance successfully avoids visual clutter when presenting multiple point clouds simultaneously. The pipeline diagram is also concise and easy to follow. In terms of content organization, the paper discusses accuracy comparisons with prior SOTA methods, which aligns with the core contribution: a unified framework for both rigid and non-rigid registration.

Significance: This work represents an important and original attempt to unify rigid and non-rigid point cloud registration within a single framework. The method not only advances conceptual understanding by bridging two traditionally separate problem formulations, but also delivers competitive accuracy across diverse benchmarks. This dual achievement—methodological unification paired with strong empirical performance—suggests the work could influence future research directions and provide practical utility for applications requiring versatile registration capabilities.

Originality:  The paper presents a commendable original contribution by leveraging a large-scale 3D model (TRELLIS) to extract local structural-semantic representations, successfully unifying rigid and non-rigid registration that are traditionally treated separately. While the current implementation inherits the extremely high computational complexity of TRELLIS—posing notable bottlenecks in runtime—this limitation does not diminish the novelty of the core conceptual integration. The framework appears particularly promising when combined with future optimizations for implicit representations, which could significantly enhance its practical applicability and broaden its impact.

---

> ### Author Rebuttal · Authors · 2026-03-30
>
> We thank the reviewer for this helpful comment.
> ### Issue 1: Failure Cases
> We agree that the current paper lacks explicit failure-case analysis and visualization.
> In the revised paper, we will add representative failure-case visualizations together with a discussion of the corresponding failure modes. In particular, we observe that RGGT fails primarily on extremely low-overlap examples (e.g., 3.58% overlap on ModelNet), where shared geometric evidence is insufficient for reliable registration. We have provided one representative example at the following anonymous link: [FailureCase.png](https://anonymous.4open.science/r/anoymize-F8C7/FailureCase1.png).
> We will include these examples and discussion in the revised manuscript to provide a more complete and balanced evaluation of robustness.
> ### Issue 2: Quantitative Analysis of Trellis Features
> We agree that the PCA analysis is qualitative and have supplemented it with a direct quantitative evaluation. Specifically, we perform nearest-neighbor matching in the Trellis feature space to obtain correspondences, and then evaluate the resulting matches using Inlier Ratio and Feature Matching Recall. On the ModelNet dataset, this procedure achieves 57.5 IR and 98.5 FMR. Notably, these results are already comparable to those of some earlier registration methods. For example, FCGF reports 56.8 IR and 97.4 FMR on ModelNet. This confirms that Trellis features alone already carry registration-relevant information. We will add these new results and revise the discussion accordingly.
> ### Issue 3: Trellis Text Conditioning
> In our current implementation, the textual prompt provides only object-level category information (i.e., the shape name), rather than any fine-grained geometric description. This is also consistent with our use of the second-stage SFT in Trellis, where the model is already strongly conditioned on the voxelized 3D shape itself, so much of the geometric information is carried by the 3D input rather than the text prompt.
>
> We have also tested Trellis without category-specific text input by replacing the original prompt with an empty prompt on ModelNet. In this setting, RGGT achieves 1.466/0.017, compared with 1.452/0.013 in the original setting, indicating only a negligible drop. This suggests that RGGT does not critically rely on textual conditioning for registration performance; instead, the text mainly acts as a weak semantic cue, while the dominant signal comes from the voxelized 3D input and the resulting Trellis prior features. We will clarify this point in the revised paper.

---

> > ### Author Rebuttal · Reviewer_nujG · 2026-04-03
> >
> > Thanks for your rebuttal. I maintain my original score.

---

> > > ### Author Response · Authors · 2026-04-04
> > >
> > > Thank you very much for your thoughtful follow-up. We sincerely appreciate your encouraging feedback and your time in considering our rebuttal. If any aspect would benefit from further clarification, we would be happy to provide additional explanation.

---

### Official Review · Reviewer_e47c · 2026-03-12

**Soundness:** 3
**Presentation:** 3
**Significance:** 2
**Originality:** 3
**Overall Recommendation:** 3
**Confidence:** 4

**Summary:**

This paper presents a unified approach to point cloud registration across rigid and non-rigid settings. The key insight is that recent 3D Foundation Models (specifically TRELLIS) can encode point clouds where the point-wise features retain both local geometric and global semantic properties.  Thus, these features contain information useful for both rigid and non-rigid registration.  The proposed RGGT model builds on this strong prior with a transformer model using global-, self-, and cross-attention to refine the TRELLIS features for correspondence. Training losses include both contrastive correspondence and reconstruction (coordinate regression) losses.

**Compliance With Llm Reviewing Policy:**

Affirmed.

**Final Justification:**

I appreciate the author rebuttal, and most of my initial questions were answered satisfactorily.

(1) I regard the key contribution of this paper is the insight that TRELLIS pt cloud embeddings/features retain both local geometric and global semantic properties, and thus a full analysis of TRELLIS (e.g. varying model sizes) would be a necessary component of the paper.

(2) As acknowledged by the authors in the rebuttal, there is no meaningful difference between the metrics using Part-Field vs Trellis.  As their targets are different (3D vs semantics), there is a need for more analysis to understand the capabilities of these foundation models.

I acknowledge that neither of the above points can be addressed in a short rebuttal window.

I will maintain my original rating of Weak Reject.

**Key Questions For Authors:**

Please see my questions in the Strengths & Weaknesses section.  Additional questions are below:

- How are the results of non-rigid methods on rigid benchmarks collected (Table 2)?   I don’t see the GraphSCNet paper reporting results on rigid benchmarks.  From a quick read of GraphSCNet it looks like prediction is a deformation graph or collection of local rigid transformations. How is the rigid transformation extracted? Same question applies to other nonrigid methods that may be reported in Table 2.
- The reported results for GraphSCNet for generalization to CAPE (Table 3) do not match the reference paper (https://arxiv.org/pdf/2303.09950, Table 3). There might be a simple explanation for this but it would be helpful if the authors could explain what has changed in the experimental setting.

**Limitations:**

There is no discussion relating to technical limitations.  There is a section on societal impact and that explanation is sufficient.

**Strengths And Weaknesses:**

**Strengths**
- Non-rigid and rigid point cloud alignment have been treated as distinct problems methodologically, to the point that very few papers (none?) show results on both rigid and non-rigid datasets.
- This paper observes an interesting property of the TRELLIS 3D foundation model, that its feature encodings retain the geometric and semantic information needed for both the rigid and non-rigid tasks.  This insight is particularly interesting because as far as I can tell TRELLIS is not trained with any articulation/deformation datasets or augmentation strategies.
- The correspondence feature refinement transformer architecture has an intuitive design, and shows impressive results across the common datasets for rigid and non-rigid registration.

**Weaknesses**
- The TRELLIS encoder requires a voxelized point cloud and text conditioning, meaning the RGGT model always needs text conditioning.   This is a limitation that other models do not need (for example many registration methods don’t require to know the object category).  This point needs to be addressed to make fair comparisons.  Perhaps there is an off-the-shelf captioning tool that can be used to provide the text conditioning for evaluation purposes?  This is particularly important for generalization experiments where one might test on point clouds that come from beyond the training categories.  If some of the other competitive baselines also utilize category information then the comparison would be fair as-is (although I still suggest making it clear which methods require this information).
- Another point regarding fair comparison is that the Implementation Details (sec 4) indicate that RGGT is trained on all 4 datasets.  The baselines being compared against do not have this benefit.  I believe the closest baseline (GraphSCNet) is only trained on 4DMatch for example.  What happens if GraphSCNet and other baselines also have the benefit of training on all datasets?
- The ablations in Table 4 tell us the impact of individual components in the transformer model relative to the full RGGT design, but we don’t get to see the impact of the proposed transformer model relative to other possible correspondence architectures.  For example, what if TRELLIS features were given as input to some baseline models, would they also reach state of the art performance?
- Further analysis of TRELLIS would be welcome. From their download page (https://github.com/Microsoft/TRELLIS) it seems multiple foundation model sizes are available ranging from 342M to 2B. It might be interesting to see RGGT performance as a function of the TRELLIS checkpoint
- TRELLIS over PartField (sec 4.6).  The performance gap swapping TRELLIS with PartField is negligible (esp. compared to the gap between RGGT and prior works).   The attribution is made to “TRELLIS’s structured geometric priors designed for 3D generation” while “PartField focuses on semantic part segmentation.”  Given the performance is essentially the same I’m not sure the conclusion is appropriate.

I look forward to seeing the author's response to the points above and discussing with the other reviewers.

---

> ### Author Rebuttal · Authors · 2026-03-30
>
> Thank you for the comment.
> ### Issue 1: Trellis Text Conditioning
> We agree that using an off-the-shelf 3D classifier or image/point-cloud captioning model to generate approximate prompts is promising, especially for open-world evaluation and unseen categories. Our additional ablation suggests that RGGT does not critically rely on explicit category text. Replacing the category prompt in Trellis with an empty prompt, RGGT still achieves 1.466 / 0.017 RRE/RTE on ModelNet, close to the full model (1.452 / 0.013) and better than the second-best baseline (2.880 / 0.028). This suggests that the main benefit comes from Trellis’s native 3D structural prior, while text conditioning acts as a mild semantic regularizer rather than a necessary source of supervision.
> We will include this empty-prompt ablation and clarify which methods use category or semantic information.
> ### Issue 2: Training Fairness
> We acknowledge this disparity and, due to time constraints, provide a fairer comparison by retraining GeoTransformer, GraphSCNet’s base architecture, on all four datasets. On ModelNet, RGGT achieves 1.452 / 0.013, versus 2.564 / 0.026 for GeoTransformer trained only on ModelNet and 2.512 / 0.022 when trained on all four datasets. When trained only on ModelNet, RGGT still achieves 1.513 / 0.015. This shows that the gain is not solely due to multi-dataset joint training.
> ### Issue 3: Transformer Architecture Comparison
> To isolate this factor, we conduct two additional experiments on ModelNet only. First, plugging Trellis features into GeoTransformer improves performance from 2.564 / 0.026 to 2.131 / 0.022, but remains worse than RGGT (1.513 / 0.015). Second, replacing the RGGT backbone with the GeoTransformer module while keeping the rest unchanged yields 1.723 / 0.019, again below full RGGT. These results suggest that the gain is not due to Trellis features alone. Instead, both the generative prior and the RGGT backbone design contribute, and their combination yields the best performance. We will add these comparisons.
> ### Issue 4: Trellis Backbone Scale
> A full comparison across Trellis scales is costly in our setup: each checkpoint requires re-extracting Trellis features for all point clouds and retraining a separate RGGT model from scratch. This process would take over one week, beyond the rebuttal timeline. We therefore provide a preliminary sensitivity analysis by replacing the Trellis-text backbone at inference while keeping the rest of RGGT fixed.
>
> On ModelNet, the public 342M and 1.1B checkpoints yield 1.974 / 0.021 and 1.764 / 0.018, versus 1.452 / 0.013 for the default 2B checkpoint. These results show a clear scale-performance trend: larger Trellis backbones perform better. RGGT remains effective with smaller checkpoints, suggesting that the method is not tied to a specific 2B model, although backbone scale matters.
> We will add this analysis and note that it is a preliminary inference-stage study.
> ### Issue 5: Partfield vs Trellis
> We agree that our attribution in Sec. 4.6 was too strong given the small gap between Trellis and PartField.
>
> A more appropriate conclusion is that both Trellis and PartField provide strong 3D priors, and Trellis shows only a slight advantage in our current setting rather than a decisive one. One possible explanation is that PartField offers reliable part-level features, while Trellis may provide denser geometry- and semantics-aware features more favorable for point-level correspondence. However, our current experiment does not support a strong causal claim.
>
> This interpretation is broadly consistent with prior diffusion-based correspondence works such as Emergent Correspondence from Image Diffusion and A Tale of Two Features: Stable Diffusion Complements DINO for Zero-Shot Semantic Correspondence, which suggest that diffusion features can encode spatially organized semantic/geometric information useful for dense matching.
> We will revise the wording to better match the evidence.
> ### Issue 6: Non-rigid Method Result on Rigid
> You are correct that methods such as GraphSCNet are not designed for rigid registration, so their results on rigid benchmarks are obtained with a unified post-processing protocol rather than taken from the original papers.
>
> For non-rigid methods, we use their final per-point predictions to obtain point-wise correspondences (or equivalently, the deformed source point cloud), and then estimate a global rigid transformation $T$ by standard SVD-based alignment. The same protocol is applied to all non-rigid methods in Table 2 to ensure a fair comparison on rigid benchmarks.
> ### Issue 7: CAPE Discrepancy
> The difference comes from the CAPE evaluation split. The GraphSCNet paper reports results on 11,288 sampled pairs from CAPE, but this split is not publicly available. For reproducibility, we re-evaluated GraphSCNet on the public CAPE test set of 2,508 pairs. Therefore, the numbers in Table 3 are obtained under a different test setting from the original paper.

---

> > ### Author Rebuttal · Reviewer_e47c · 2026-04-03
> >
> > Thanks for the detailed response to each listed question.  Although there are some outstanding experimental questions that cannot be answered in a short rebuttal period (as acknowledged in the rebuttal), I do have enough information now to finalize the review.

---

> > > ### Author Response · Authors · 2026-04-04
> > >
> > > Thank you very much for your thoughtful follow-up. We sincerely appreciate your careful reading and constructive feedback throughout the review process. We will incorporate these additional results and clarifications into the future version. Please also let us know if there are any remaining questions or issues that you would like us to clarify further. Thank you again for your time and support.

---

### Decision · Program_Chairs · 2026-04-30

**Decision:**

Accept (regular)

**Comment:**

All reviewers appreciate the novelty of the approach of using Trellis features for point cloud rigid and non rigid registration. They raised concerns regarding the text prompt of the model, regarding marginal improvement over partfield, and regarding lack of failure and  runtime analysis. The rebuttal submitted by the authors addressed these concerns to some extent. Overall, the reviewers and the AC find this a useful contribution due to its novelty and good empirical performance. The authors are requested to improve and revise the paper as per the discussions.